# German Language Adaptation of the NAVS (NAVS-G) and of the NAT (NAT-G): Testing Grammar in Aphasia

**DOI:** 10.3390/brainsci11040474

**Published:** 2021-04-08

**Authors:** Ruth Ditges, Elena Barbieri, Cynthia K. Thompson, Sandra Weintraub, Cornelius Weiller, Marek-Marsel Mesulam, Dorothee Kümmerer, Nils Schröter, Mariacristina Musso

**Affiliations:** 1Department of Neurology, University Medical Center Freiburg, Breisacherstrasse 64, 79106 Freiburg, Germany; ruth.ditges@googlemail.com (R.D.); cornelius.weiller@uniklinik-freiburg.de (C.W.); dorothee.kuemmerer@uniklinik-freiburg.de (D.K.); nils.schroeter@uniklinik-freiburg.de (N.S.); 2Department of Communication Sciences and Disorders, Northwestern University, Evanston, IL 60208, USA; elena.barbieri@northwestern.edu (E.B.); ckthom@northwestern.edu (C.K.T.); 3Mesulam Center for Cognitive Neurology and Alzheimer’s Disease, Feinberg School of Medicine, Northwestern University, Chicago, IL 60611, USA; sweintraub@northwestern.edu (S.W.); mmesulam@northwestern.edu (M.-M.M.); 4Department of Neurology, Northwestern University, Chicago, IL 60611, USA; 5BrainLinks-BrainTools Excellence Cluster, University of Freiburg, 79106 Freiburg, Germany

**Keywords:** syntactic competence, aphasia, aphasia’s therapy

## Abstract

Grammar provides the framework for understanding and producing language. In aphasia, an acquired language disorder, grammatical deficits are diversified and widespread. However, the few assessments for testing grammar in the German language do not consider current linguistic, psycholinguistic, and functional imaging data, which have been shown to be crucial for effective treatment. This study developed German language versions of the Northwestern Assessment of Verbs and Sentences (NAVS-G) and the Northwestern Anagram Test (NAT-G) to examine comprehension and production of verbs, controlling for the number and optionality of verb arguments, and sentences with increasing syntactic complexity. The NAVS-G and NAT-G were tested in 27 healthy participants, 15 right hemispheric stroke patients without aphasia, and 15 stroke patients with mild to residual aphasia. Participants without aphasia showed near-perfect performance, with the exception of (object) relative sentences, where accuracy was associated with educational level. In each patient with aphasia, deficits in more than one subtest were observed. The within and between population-groups logistic mixed regression analyses identified significant impairments in processing syntactic complexity at the verb and sentence levels. These findings indicate that the NAVS-G and NAT-G have potential for testing grammatical competence in (German) stroke patients.

## 1. Introduction

A speaker’s ability to use a language requires mapping between an unlimited number of thoughts and meanings, and an unlimited number of words and associated sound sequences [1,2]. Grammar is the set of structural rules governing this mapping. From the early work of computational linguistics to the present day [3,4,5] it has been suggested that the critical aspect of linguistic ability is the existence of this predetermined rule system. Thus, syntactic knowledge is crucial for the use of language. Notably, however, syntactic competence relates to organism-external factors including ecological, cultural, and social environments, as well as general sensory–motor operations (language perception and production) and conceptual–intentional computations, such as (focused) attention and working memory, in order to form thoughts and meaning [6,7].

From linguistic and psycholinguistic data, it is well known that verbs are lexical items that possess syntactic argument valence, constituting one of the basic building blocks of grammatical clauses [6,7,8,9,10,11]. Argument valence sets the structure of arguments and expresses the relationships among nouns in various thematic roles (see the Projection Principle and the theta criterion [9,11]). Each verb-argument assigns a specific so-called thematic role, for example, the agent (corresponding to the noun phrase (NP) in subject position), the theme (e.g., NP in object position that is affected by the action), and the goal that the action leads to [12]. For example, “to put” is a verb with three obligatory arguments and thus requires at least three participant (thematic) roles, i.e., agent, theme, and goal. A sentence requires the correct expression of all these arguments as in, for example, “The man (agent) is putting the box (theme) on the shelf (goal).”

At the sentence level, the notion of syntactic complexity has received considerable attention [13,14,15,16,17,18]. One aspect of increased complexity within a sentence relates to the number of phrasal nodes—e.g., the building blocks of a sentence that are dependent on each other in a tree structure and whose syntactic movements are necessary for building a sentence [13,19,20,21]. The greater the number of phrasal nodes a unit dominates within a sentence, the more complex the sentence is. Therefore, nesting or center-embedding sentences, such as relative clauses, are more difficult than active sentences [1,22].

However, it is well known that there are differences in complexity between sentences that have the same number of embedded elements, as between subject relatives (SR: as “Pete saw the man who is saving the woman”) and object relatives (OR as “Pete saw the woman who the man is saving”), in that the latter involve long-distance dependencies [23,24,25]; that is, verb arguments (i.e., agent, theme) are nonlinearly entered into sentences, so that the theme precedes the agent and the two are separated by intervening linguistic material. According to the Dependency Locality Theory [25,26], the distance over which these dependencies are computed forms a hierarchy of sentence complexity [23,24,25,27]. In this light, OR sentences are more complex than SR sentences because the processing cost of integrating verb arguments is proportional to the distance between their presentation and their resolution, a process which relies on working memory [27].

Over the past thirty years, behavioral and functional imaging studies have confirmed linguistic claims concerning syntactic complexity in healthy children, in children with syntactic language impairments [24,28,29] and in healthy participants, both behaviorally and using functional imaging [30,31,32,33]. Syntactic complexity also affects verb and sentence processing in stroke-induced aphasia—an acquired language disorder often resulting from lesions within the language network [34,35,36,37]. Studies show that patients with Broca’s aphasia and concomitant agrammatism are particularly affected by syntactic complexity [8,34,37,38]. However, several studies have also documented difficulties in comprehension and production of verbs and sentences with increasing complexity in patients with anomic or Wernicke’s aphasia, indicating that such deficits are not limited to Broca’s aphasia [8,36,38,39,40]. In the same way that selective lesions of Broca’s region can induce mild, transient aphasia but not agrammatism [40], patients with lesions outside of Broca’s area may show grammatical impairments [38,39,40]. Functional imaging data show that, on the one hand, Broca’s area continues to be a critical brain region for syntax processing and acquisition [41,42,43,44]. On the other hand, this area does not work alone—Broca’s area operates together with other left perisylvian regions with which it interacts via dorsal and ventral pathways—nor does it operate exclusively in the language domain [42,44,45]. Current scientific opinion accepts that syntactic mapping is both localized and distributed [46,47]. These findings provide important evidence in favor of testing patients who present aphasia resulting from disparate lesions for verb and sentence processing impairments [7,32,34,35,39,48,49,50,51,52,53,54].

Indeed, the validity of aphasic syndromes based on the Wernicke and Lichtheim anatomical model of language [55] is being increasingly questioned [34,38,56,57,58,59,60,61,62,63,64,65] and neuropsychological tests, which are based on this model and lead to a diagnosis of aphasia (and type of aphasia), are inadequate for understanding impaired language processes. Current approaches to aphasia assessment seek to analyze patients with regard to linguistic and psycholinguistic models of grammar to highlight specific language impairments and treatments [34,66,67,68,69,70,71,72,73,74,75,76]. There is a large consensus that, first, only when syntactic deficits are recognized is it also possible to treat them. Second, in general, protocols that exploit the linguistic and psycholinguistic properties of sentences result in more effective treatment and generalization effects than direct production approaches that consider only the surface form of target sentences (e.g., [77]). Third, several studies agree that syntactic complexity is an overarching principle of treatment for grammatical disorders, as treatments which focus on more complex forms result in generalization to fewer complex structures [66,67,78,79,80].

The standard diagnostic battery for chronic aphasia in Germany is the Aachener Aphasia Test (AAT) [57,58,59,60,61,62,64,65]. This test evaluates spontaneous speech and includes five subtests to assess a speaker’s language functioning across repetition, naming, comprehension, reading, and writing modalities, but not syntax [60,61,62,64,65].

Only three tests are available to examine syntactic abilities in the German language: the “Komplexe Sätze” [81], “Sätze verstehen” [82] and “Sprachsystematisches Aphasiescreening (SAPS)” [83] (see Appendix A for an overview of aphasia tests in the German language). However, none of these tests allows—in one test battery—an in-depth and comprehensive examination of grammar, including tests for verbs and verb argument structure, and assessment of comprehension and production of various sentence types.

Importantly, several neuropsychological studies have found that grammatical disorders at the verb level are frequent in aphasia and that such deficits are affected both by the number and the type of arguments selected by verbs ([8,84,85,86,87,88,89]; see Thompson and Meltzer-Asscher [32] for a review). In addition, it has also been demonstrated that the obligatory (Ob) or optional (Op) status of verbal arguments—i.e., whether a verb requires the arguments involved in its lexical representation to be present in the grammar or allows some arguments to be omitted (i.e., they are optional)—may contribute to verb and sentence deficits in aphasia [8,89,90]. Training verb deficits, emphasizing argument structure and thematic role mapping, effectively improves verb and sentence production; and results in the recruitment of neural networks relevant for verb and argument structure processing in healthy individuals [91]. However, current German batteries for testing grammar do not test verb processing; rather they examine only sentence-level abilities.

Tests of sentence processing in German also bear some major limitations. Namely, “Komplexe Sätze” (in English, complex sentences) tests sentence production only, whereas “Sätze verstehen” (sentence comprehension) examines sentence comprehension only [81,82]. In addition, the former [81] only includes non-canonical sentences (i.e., sentences in which the order of constituents is different from the standard Subject-Verb-Object (S-V-O) order), whereas the latter assesses only active (reversible and non-reversible) sentences, and SR and OR structures [82]. Thus, available tests for evaluating sentence processing in German are limited. First, it has been shown that patients’ performance in comprehension and production may be dissociated (e.g., [92,93,94,95]), and treatment studies indicate that treatment provided for sentence comprehension does not generalize to production (or vice versa; [66,96,97]). Thus, it is important to test the same sentence types across domains. Second, although non-canonical sentences are typically more difficult for aphasic participants than canonical sentences [35,53,92,98], patients are often impaired even on syntactically simple sentences (e.g., active sentences [99]; see [100]). Thus, any test for syntactic processing needs to evaluate both simple and complex sentences. Third, according to Chomsky’s Principles and Parameters theory [9], there are several distinct types of complex sentences: those with noun phrase (NP)-movement, involving movement of a noun phrase (NP) from an argument position to another argument position as in passive sentences, and those with Wh-movement, involving movement of an argument to a non-argument position as in ORs. Notably, in aphasia, training sentences with Wh-movement (e.g., OR structures) does not influence the production or comprehension of sentences with NP-movement, such as passives (and vice versa), although generalization across sentences with similar movement operations is commonly seen (e.g., from object relative structures to object Wh-question forms) [67,69,72,73,74,75,76]. Therefore, a complete assessment of syntactic abilities in aphasia requires testing different types of complex sentences.

Finally, the SAPS is a relatively novel test enabling a detailed analysis of a patient’s phonological, lexical, and morphosyntactic abilities, including the comprehension and production of complex sentences. However, in addition to the lack of verb production and comprehension assessments, sentence comprehension is examined only for Agent-first and Agent-second structures, while sentence production is assessed only for Object Verb and Subject, Verb, Object clauses. Further to this, NP- and Wh-movement are not systematically tested.

In summary, existing tests fail to provide a precise picture of the aphasic syntactic deficits. Verbs are not classified and compared in terms of the number and optimality of an argument structure, and canonical versus non-canonical contrasts are not explicitly addressed. Moreover, the stimuli included do not control for frequency of occurrence, despite this having a strong influence on a patient’s language processing [101]. Overall, these findings highlight the need for reliable tools in research and clinical work in order to test grammar in the German language.

A recently developed test, the Northwestern Assessment of Verbs and Sentences (NAVS) [8], is very promising in terms of both design and results. The NAVS includes tests for (1) verb naming (i.e., the Verb Naming Test, VNT)—controlling for frequency as well as the number (one, two, and three) and optionality (obligatory vs. optional) of verb arguments; (2) verb comprehension (i.e., the Verb Comprehension Test, VCT), which uses the same items tested in the VNT; (3) argument structure production (i.e., the Argument Structure Production Test, ASPT), which examines the effects of argument number and optionality on the production of active sentences with the same target verbs tested in the VNT; and lastly, (4) comprehension and production of canonical and non-canonical sentences with increasing syntactic complexity, which are associated with NP- and Wh-movement, i.e., the Sentence Comprehension Test (SCT) and the Sentence Production Priming Test (SPPT), respectively. Thus, the NAVS allows for an analysis of both comprehension and production of verbs and sentences in the same test battery and uses the same verbs and sentence types across modalities. The results of the original version have shown that NAVS is sensitive to capturing syntactic processing in English-speaking people with aphasia [8]. Currently, there are Italian and Chinese versions of the NAVS (NAVS-I [7] and NAVS-C [36], respectively). All versions have confirmed verb deficit and impairment at the sentence level in both comprehension and production, reflecting syntactic complexity hierarchies in patients with aphasia.

The Northwestern Anagram Test (NAT) [102] is another English test for sentence production. Modelled after the NAVS, it examines the same sentence structures as the NAVS, but rather than requiring participants to produce sentences, it requires the arrangement of word cards to form sentences. Therefore, it is able to bypass severe motor speech deficits and word-finding difficulties, as well as avoid potential interference from working memory deficits [103]. The NAT was developed to test patients with primary progressive aphasia [104], which, in contrast to stroke-induced aphasia, is caused by a degenerative, progressive disease. The results of the original version [102], and the Italian Version NAT-I [23], have demonstrated a high correlation with the NAVS SPPT and a canonicity effect (i.e., better accuracy for canonical than non-canonical sentences), even though it is not as sensitive as the NAVS with regards to NP- and Wh-movement. Thus, NAT cannot function as a substitute for a thorough investigation of grammatical processing; nevertheless, it appears to be a valid instrument that can be used for evaluating some patients to identify basic syntactic ability.

The current study aimed to develop German versions of the NAVS and NAT, hereafter, the NAVS-G and NAT-G, respectively, and to test the “feasibility” of these tests for assessment of grammar deficits in German patients with language impairments. The main idea was to take the first step toward developing a flexible novel tool, which, depending on the patient’s clinical characteristics and therapeutic options, might include the NAVS-G or a combination of VNT, VCT, and ASPT of the NAVS-G and the NAT-G. The validity of a neuropsychological test—which is a requirement to justify its use and interpretation of scores—is established through the collection of several kinds of evidence, and by testing a large sample of participants [105,106,107]. Rohde et al. [107], in a systematic search of 161 publisher databases (including the AAT), did not find a single diagnostic test for aphasia in stroke populations that had sufficient evidence of validity. Before embarking on such a large study, we aimed to test, in a relatively small sample, whether the German adaptations of NAVS and NAT have the potential to elucidate grammatical competence in (German) stroke patients with aphasia.

Given the centrality of grammar for language processing, and the frequency of grammatical deficits in patients with aphasia, we tested the sensitivity of the German adaptations of the NAVS and NAT to detect grammatical deficits at the verb or the sentence level, in patients with mild or even “residual” aphasia. Characterizing residual language deficits in mild forms of aphasia is often challenging, due to the lack of diagnostic tests that are sensitive to aspects of verb and sentence complexity that are known to affect both production and comprehension (see above). Thus, detecting grammatical impairments in individuals with residual aphasia would ensure that the NAVS-G and NAT-G are sufficiently sensitive to be used with individuals with frank agrammatism or anomia, as shown for the original versions and for their adaptations [7,8,36,102].

The first aim of this study was to replicate the results obtained with the English version regarding the effects of verb-argument structure complexity (i.e., better accuracy for one-argument than for two- and three-argument verbs) and canonicity (i.e., better accuracy for canonical than for non-canonical sentences), in German patients with aphasia. Therefore, we adapted the NAVS and NAT to the German language, by taking into account the linguistic differences between both languages (German and English; for details see Method), and administered them to 15 left hemispheric stroke patients (LHSP) with a well-documented history of aphasia and who had stable, mild to minimal language impairments at the time of testing. For controls, we tested the NAVS-G and NAT-G, not only in healthy participants (HP) [24], but also in right hemisphere stroke patients (RHSP) devoid of language impairments [108]. The inclusion of a RHSP group, which was well-matched to the aphasia group in terms of demographic and clinical data, allowed us to control for the effect that ischemic brain lesions have on test performance [109]. Moreover, it has been shown that, despite the left hemisphere dominance for language [110,111,112,113], RHSP may have difficulties in syntactic processing [114,115]. If lesions to the right hemisphere affect grammatical task performance [42], it is crucial to identify which tasks of the NAVS-G and NAT-G may detect syntactic deficits in stroke patients who had similar lesion localization and extension to the patients with aphasia, but in the right hemisphere.

Notably, it has been shown that demographic factors such as age [116] and education [117], and clinical variables such as lesion location, size, etiology (hemorrhagic vs. ischemic), stroke severity, and the presence of a new stroke, are factors that can predict language ability in patients with aphasia [118,119,120,121]. In light of this evidence, we controlled for several of these factors, both with our study protocol and in our data analyses. A second aim of the study was, therefore, to identify how NAVS-G and NAT-G performance might relate to demographics and clinical variables as a prerequisite for further, more extensive studies of test standardization and validation.

## 2. Materials and Methods

### 2.1. Study Design

A critical concept for neuropsychological assessment is to control the influence of sociodemographic factors on test performance [122,123]. In accordance with international clinical guidelines for stroke-induced rehabilitation [124,125], this study protocol provided and collected detailed clinical data using appropriate instrumental diagnostics and validated assessment batteries, as they are crucial for test interpretation (in stroke patients). Participants’ age, gender and education, as well as data reflecting stroke severity—using the National Institute for Health Stroke Scale (NIHSS) [126], the modified Rankin Scale (mRS) [127] and the Barthel Index (BI) [128]—were collected (see Table 1). The majority of these scores were extracted from the institutional research database of the Freiburg clinic (Large Scale Project), from which most of the patients participating in the study were selected.

Inclusionary criteria for all participants were: (1) age between 18 and 90 years, (2) German as a native language, (3) right-handedness [129], and (4) sufficient cognitive abilities to comply with the test requirements and provide informed consent. The exclusion criteria included: (1) history or current diagnoses of other medical, neurological, or psychiatric disorders interfering with participation or data analysis, (2) severe hearing or vision deficits, (3) early bilingualism [130], and (4) music education at the professional level [44,131,132].

To ensure comparisons between well-matched groups (see [107,109]), stroke participants were included only if they suffered from a first ever-ischemic lesion of the arteria cerebral media (middle cerebral artery) in the left or the right hemisphere [118,120,121]; patients with bilateral or hemorrhagic strokes were excluded [119]. Moreover, as the target population was the stroke population, we also excluded patients with other structural brain or skull lesions that were reported or were visible on the MRI/computer tomography. Thus, each patient underwent a neuroimaging diagnostic exam (CT or structural MRI). Structural MRI was preferred, as it allows precise mapping of stroke lesions. Three left hemispheric stroke patients and four right hemispheric stroke patients had no MRI scans. MRI scans were obtained on a 3T Trio scanner (Siemens) on an average of 1.9 days after symptom onset (SD 2.8, min 0, max nine days). Clinical and imaging data were entered as variables in statistical analyses and were used for between-groups comparisons.

Spontaneous speech samples were collected from all stroke participants (both left- and right-hemisphere brain-damaged, i.e., LHSP and RHSP) to evaluate overall communicative abilities. These samples were collected, according to the ATT guidelines, via a structured interview covering four conversational topics (onset of illness, profession, family, and hobbies) of about 10–20 min. They were then evaluated based on six dimensions of linguistic behavior, with their performance on each dimension being rated on a six-point scale (see Table 1). The scores (0–5 on each level) are defined by specific and detailed criteria and require an expert speech therapist to assign them. Notably, spontaneous speech analysis is a primary method for detecting aphasia and quantifying language impairments in patients in the acute phase [133,134], or in patients with minimal deficits [135,136].

The LHSP were administered the Token Test subtest of the ATT, which is also well standardized and highly sensitive for the diagnosis of aphasia [137,138,139]. Following ATT guidelines, the binary diagnosis (aphasia yes/no) was determined by combining spontaneous speech scores with those from the Token Test subtest, and, in some cases, written language subtest [137]. To evaluate the four standard aphasia syndromes, the participants were also administered the remainder of the ATT ([137], for details see [65]), as the combination of the AAT spontaneous speech and TT (and written language) subtests is sufficient for the classification to the “no/residual aphasia” category only. Aphasia severity is based on stanine norm data available for each AAT subtest (see AAT manual). To summarize, the AAT’s spontaneous speech analysis and the Token Test (and when indicated in the AAT manual, also the subtest writing) were used in order to diagnose aphasia in all the LHSP, while the complete AAT was administered to more impaired patients for aphasia assignment. AAT’s interview and the AAT’s subsets were administered at two time points: in the acute phase (1–3 days after stroke) and in the chronic phase, i.e., at NAVS-G/NAT-G testing time (3.05 years after stroke, range: 5 months—25.8 years). Reporting the individual longitudinal aphasia symptoms is an essential aspect when studying patients with residual aphasia [137,138].

For some stroke patients who were part of the large-scale protocol study in Freiburg, it was possible to report the MoCA (Montreal Cognitive Assessment) test scores [63] in the follow-up phase. This test is very sensitive for testing general cognitive abilities, even in patients with anomic deficits [140].

The local ethics committee approved this study.

### 2.2. Participants

After excluding a woman from the HP group because of a reported dementia diagnosis, a total of 57 participants were included in this study: 15 with left-hemisphere (LH) stroke-induced chronic aphasia (eight males, six females; mean 61.35 age ± 8.18 years, range 48–71) and 42 without aphasia—the control group (19 males, 20 females; mean age 59.9 years, ±range 25–88). The control group included 27 HP (13 males, 14 females, mean age 55.9 ± 18.7 years, range 25–88) and 15 right hemisphere stroke participants (RHSP) (seven male, eight female, mean age 68.2 years, range 31–88). All participants met the inclusionary and exclusionary criteria. All participants provided written informed consent before participation.

Aphasic (LH) and non-aphasic participants (HP and RHSP) did not differ in age (t(55) = 0.332, Z = −0.332, *p* = 0.741, by normal distribution—Kolmogorov–Smirnov: *p* = 0.200 for patients with aphasia and controls), gender (*p* = 0.46. Z = −0.739, Mann-Whitney-U-Test), or education (*p* = 0.536, Z = −0.619, and *p* = 0.906, Z = −0.118, Mann-Whitney-U-Test for school leaving certificate and highest educational qualification, respectively). Similarly, no differences were found between HP and LH participants for age (t(40) = 1.159, *p* = 0.253, Z = −0.841), gender (Z = −0.318, *p* = 0.796 (Mann-Whitney-U-Test)) or education (Z = −0.469, *p* = 0.674, and Z = −1.035, *p* = 0.362, Mann-Whitney-U-Test for school leaving certificate and highest educational qualification, respectively). The two patient groups (LH and RHSP) did not differ in age (t(28) = −1.428, *p* = 0.164), lesion size (Z = −1.302, *p* = 0.220), stroke severity measured by NIHSS (Z = −1.287, *p* = 0.233), mRS (Z = −1.646, *p* = 0.148) or BI (Z = −1.46. *p* = 0.233). Only the testing time after stroke was longer for LH compared to RHSP (Z = −2682, *p* = 0.007, see Table 1). Figure 1 shows the main normalized lesion of the overlap of the binarized lesions in the left and right hemispheric stroke patients, indicating that the stroke in both patient groups damaged the same territory.

The left hemispheric stroke patients had a well-documented history of aphasia (as we studied the patients longitudinally; see Appendix A) and, at testing time (3.05 years after stroke, range: 5 months—25.8 years), stable, middle to minimal language impairments (Table 1). During the acute phase, six patients of this study could not perform the AAT (not even the Token test, Appendix A) because of the severity of aphasia (five patients), and one because of clinical instability. In the chronic phase, we were unable to obtain the Token Test data of three patients (see Table 1): one patient refused further testing and, for patients number 13 and 14, the AAT’s testing was not available. The combination of the AAT spontaneous speech and Token test (and written language) subtests was sufficient for the classification to “no/residual aphasia“ for nine participants. This classification means that the patients’ clinical pictures did not suggest a specific syndrome, but instead a general language impairment that was evident in functional communication [141] in the form of phonological or semantic paraphasias, which are syntactic or word-finding disorders (Table 1). In line with this, they also showed difficulties in naming, sentence repetition, verbal fluency, and verbal memory in the MoCA test (see Appendix A) [63,140].

The complete AAT was administered to six further impaired patients, and the respective aphasia syndrome was reported (Table 1 and Appendix A). Table 1 shows that the aphasia group’s language profile included patients with different types of chronic aphasia (from anomic to Broca’s aphasia) and severity (from minimal to mild).

Syntactic difficulties were, to some extent, detected by the AAT rating of syntactic structure in spontaneous speech based on the interview: eight patients had the maximal scores, while the others obtained lower scores due to production of short or very short sentences, and to omission/substitution of verb inflections.

### 2.3. Adaptation of NAVS and NAT to the German Language

We designed the NAVS and NAT versions for the German language. The typological differences between English and the German language dictated the need for an adaptation, and not a mere translation, of the test. The English language has a more fixed word order, maintains the SVO (subject-verb-object) construction in the main and dependent clauses whilst using the SOV order in non-canonical clauses. The basic word order of German is generally assumed to be subject-object-verb (SOV; e.g., [143]); however, due to the verb-second requirement, the finite verb must raise to the complementizer position in German main clauses (e.g., [144]), and as a result, precedes its object in neutral declarative clauses. The past participle (in the passive construction, for example), but not the auxiliary verb, must always be the last element in the independent clause, whilst in the dependent clause, the main verb must be the last element (“Peter sah den Jungen, der die Frau küsst”: Peter saw the boy, who kisses the girl). In contrast, Wh-questions work in much the same way as they do in English and, similar to English, German has Wh- and NP-movement structures.

Verbs: To be as faithful as possible to the original version, we maintained the same unequal distribution: 5 one-argument, 10 two-argument (5 Ob and 5 Op), and 7 three-argument (2 Ob and 5 Op) verbs. Whenever possible, we retained the original verbs and performed a literal translation [8,102]. As in the original NAVS version [8], all verbs used in the subtests (VNT, ASPT, and SPPT) were the same as in the VCT. To maintain this principle, as we aimed to integrate the NAVS and NAT into one German test, we used the same verbs for both tests. All the NAT verbs [102]) could be translated to German, except “to watch,” whose German translation bears a different argument structure than the original. Three VNT verbs were replaced with verbs from the original NAT. Other VNT verbs were replaced because German translations have a different argument structure (n = 3) or are compound verbs (n = 2) with a separable prefix, which generally moves to the end of the sentence. These were used as the distractors from the target stimulus, using the same argument structure as in the English version of the VNT (for details, see the list of target verbs of the original NAVS and NAT, and NAVS-G and NAT-G in Appendix A) [8,102]. We controlled for verb-frequency using the CELEX ([145] database; available online http://celex.mpi.n (accessed on 7 January 2020) see Appendix A). Obligatory and optional verbs were equated for the log_10_ lemma frequency 1.81 and 1.85, respectively. There was no significant difference between the two groups (*p* = 0.180). Additionally, there was no significant difference between the frequency of the obligatory one-argument verbs and two-argument verbs (Ob: Z = −1.048, *p* = 0.295, and Op: Z = −0.94. *p* = 0.347) or between obligatory one-argument and optional three-argument verbs (Z = −1.936, *p* = 0.095, Mann-Whitney-U-Test). However, a significant difference was noted between the less frequent one-argument verbs and the most frequent Ob three-argument verbs (Z = −2.193, *p* = 0.028).

Nouns: We attempted to keep the same nouns that were used in the original versions. However, we used other words to create novel sentences (n = 5): Brief (letter), Brötchen (burger), Besucher (visitor), Bild (painting), Boden (floor) (see Appendix A).

Syntactic structure: From the original NAVS and NAT, we kept active (A) and passive (P) sentences, subject (SWQ) and object Wh-questions (OWQ), subject (SR) and object (OR) relatives. In the NAT-G, subject and object cleft sentences (e.g., it is the man who is saving the woman), which are part of the original version, were not used because these sentence types are uncommon in the German language.

Picture Stimuli: The drawings from the original NAVS and NAT were used as stimuli. In the NAT-G for the SR and OR sentence, the image of “Peter” was added to the original stimuli (see Appendix A).

The arrangement of the stimuli: In the original version, the stimuli were randomly arranged. Considering the possible development of a short version, we arranged the stimuli in five blocks, each including one stimulus for each category, i.e., a verb for each—Ob and Op—verb argument and a sentence of each clause types: A, P, SR, OR, SWQ, and OWQ, in random order.

The NAVS-G and NAT-G use the same structure (see also Table 2) and picture materials as in the original version. In detail, NAVS-G included:

VNT: The stimuli consisted of 22 black and white line drawings, which described the action of each verb. Participants had to name the verb describing the action of the picture.

VCT: The stimuli included 22 cards depicting the target verb (the same as in the VNT) and three distractors. One of these was the same verb type as the target, and the other two had a different argument structure. Distractors were selected from 25 additional verbs (6 one-argument, 14 two-argument, and 5 three-argument verbs). The position of the target verb picture was counterbalanced across the stimuli. The examiner pronounced the target verb, and participants had to indicate the correct one from the four displayed on the card.

ASPT: For the ASPT, 9 animate and 13 inanimate nouns were combined with the verbs used for the VNT. Each verb was tested in all of its argument structure contexts, resulting in 32 target sentences. Pictures displaying verb actions were the same as for the VNT; however, the ASPT pictures included labels with the names of the action and its participants to bypass word retrieval difficulties. Participants were required to produce the correctly inflected verb and all its arguments in the correct order using an active transitive sentence.

SPPT: This test examined the ability to produce six sentence types (n = 5 for each type): actives, SWQ, SR, (i.e., canonical sentences); and passives, OWQ, and OR (i.e., non-canonical sentences). Three examples allowed the examiner to clearly explain the task. For each target sentence, a semantically reversed counterpart was used as a prime. Thirty-two black and white line drawings depicting two actors and one action displayed the prime event on the left of the card with the target event. The semantically reversed version displayed these on the right of the card. Printed words and arrows labeled each actor and the action in the picture, in order to avoid word retrieval difficulties. For relative clauses, a man—Peter—was shown, looking over the transitive action; participants had to produce the same sentence type as the priming sentence. Participants were asked to produce a sentence describing the target picture, using the same sentence structure as in the prime sentence.

SCT: This test used the same material as the SPPT. The examiner read the target sentence aloud, and participants were asked to point to the picture corresponding to the sentence (sentence–picture matching).

NAT-G: This test assessed the production of the same sentence types as the SPPT, using the same picture stimuli. For each target sentence, cards for all the elements in the target sentence (nouns, articles, verbs, prepositions) constituting the correct sentence were printed and placed in random order under the picture by the examiner. Participants had to build a sentence to describe the picture using all the word cards. The examiner provided the first words to elicit production of the target sentence types.

Test administration followed the procedures described in the original versions of NAVS and NAT manuals. For more details, see Appendix A.

### 2.4. Scoring

For the VCT and the SCT, the examiner circled the number corresponding to the picture selected by the patient on the response forms. Each correct answer equaled one point. All correct answers were totaled to give one final score.

In the VNT, semantic paraphasias for target verbs were recorded. However, they were rated as correct provided that they included the same argument structure and the same meaning as the target verb. Phonological paraphasia and dysarthric errors were rated as correct as long as the target word was still recognizable. The form of the verb did not matter and all tenses and inflections were accepted. Each correct answer equaled one point.

For full ASPT/SPPT/NAT-G credit, the sentences were required to include the verb (correctly inflected) and all given arguments in the correct order. Omissions and substitutions of articles or pronouns did not affect scoring. Sentence fragments (e.g., the man… woman) and omissions of the verb or any of its arguments were considered errors. In ASPT and SPPT, the utterance produced was required to be free of neologisms. In SPPT, role reversal was considered an error (for a detailed description, please see NAVS and NAT manuals).

### 2.5. Data Analysis

For descriptive statistics, the response accuracy (consisting of the mean percent of correct production or comprehension of verbs or sentences) and standard deviation were calculated separately for each participant group. Participant accuracy on all tasks was analyzed in R (version 4.2, R Core Team, 2020) using logistic mixed effects regression [146].

Random intercepts for participants and items were entered in all models. Categorical variables with 3 or more levels were simple coded in all analyses.

For within-group analyses, models included the main effect of interest as the fixed effect, more specifically, for NAVS-G subtests investigating verb processing (the VNT and the ASPT) verb argument number (VAN), verb argument optionality (VAO), and verb type (VT), were entered in separate analyses as fixed factors.

To evaluate the effect of the number of verb arguments, optional and obligatory two-arguments verbs were grouped together, as were optional and obligatory three-arguments verbs (as in the original work by Cho-Reyes and Thompson [8]).

To examine the effect of argument optionality, all obligatory verbs (one-, two-, and three-argument) were grouped together and compared to all optional (two- and three-argument) verbs.

A third analysis was carried out using verb type as a fixed effect with 5 levels (Ob 1-arg, Ob 2-arg, Ob 3-arg, Op 2-arg and Op 3-arg). For the VCT and ASPT, analyses were conducted the same way as in the VNT.

For subtests investigating sentence processing (the SPPT, NAT and SCT), canonicity and sentence type were entered in separate analyses as fixed factors. To assess the canonicity effect, we compared data from passive, OWQ and OR (sentence types with non-canonical order) to active, SWQ, and SR (sentence types with canonical order). In order to check if the effect of canonicity was limited to the comparison OR > SR (as described in HP [1,11], or whether this effect extends to other sentence types, we compared data from passive and OWQ (non-canonical order) to active and SWQ (canonical order). A third analysis was then conducted using sentence type as the main predictor of interest, with six levels.

Based on the evidence that participants’ clinical condition and demographic variables may affect task performance, we included the variables age, educational graduation (Ed.g.) and qualification (Ed.q.), AAT spontaneous speech (AAT.ss) (a measure of functional communication [141]), AAT spontaneous speech syntax score (AAT.sy), i.e., the score of the subtest 6, reflecting the syntactic structure of narrative production [141], AAT global score, aphasia severity score (Aph.se), and NIHSS, BI, and mRS scores, as individual fixed effects. The covariates were entered one at a time and their contribution to the overall model fit was evaluated by comparing models with and without each variable, using the anova function included in the lme4 package [147]. Thus, the best predictors were used to refit the single models: VAN and VT for VNT and ASPT; and canonicity and sentence types for SPPT, NAT and SCT.

For between-group analyses, models included group as a main effect and interaction with the effects of interest (e.g., the same as for the respective within group analysis).

Statistical significance for fixed effects was determined by computing ps using Satterthwaite approximation, as provided within the lmerTest package [148]. For variables with three or more levels, Chi-squared statistics and *p*-values were derived from model comparisons (i.e., by comparing the model containing that variable to the model without that variable). Z-values and *p*-values are provided for continuous variables or categorical variables with only two levels.

Pairwise comparisons were carried out using the multcomp package [149], applying FDR correction for multiple comparisons.

All the reported *p*-values indicated significance levels after performing correction for multiple comparisons. In the presence of significant effects, R^2^ values for the best-fit model (fixed effects only) were computed using the r.squared GLMM function within the MuMIn package [150]. The coefficient of determination was calculated, adjusted for comparisons and reported for each model. A R^2^ was interpreted in three categories according to the guidelines: weak (R^2^ = 0.02), moderate (R^2^ = 0.13) and strong (R^2^ = 0.26) [151]. We also reported the regression estimates for capturing the impact of the explanatory variables, and the standard error of the regression estimate as goodness of model fitting [152]. In addition, power calculations were run on the best-fit model, to assess the ability of the model to detect the fixed effects, following Brysbaert and Stevens [153], and using the powerSim function provided within the simr package (see [154]) with N = 200 Monte Carlo simulations. Models yielding an observed power of at least 80% were considered sufficiently powered.

Lastly, we calculated the comparisons between subtests in LHSP with aphasia using Wilcoxon tests, due to insufficient variability in some of the subtests (in which performance was at ceiling), which would have prevented adequate model fit if using mixed-effects logistic regression. Correlations between subtests in the LHSP group were carried out using the Pearson correlation. The correlation coefficient r was interpreted in three categories according to the guidelines of Cohen [151]: light (|r| = 0.10), moderate (|r| = 0.30) and strong (|r| = 0.50).

## 3. Results

Using logistic mixed-effects regressions, we calculated the random effects for items and participants, and the fixed effects of many models (single models and adjusted models). Due to the vast amount of data generated from these analyses, we decided to report statistics for fixed effects in the manuscript, and to provide statistics for random effects in the Appendix A. For the same reason, in the manuscript we provide only significant effects and pairwise comparisons for adjusted models. All other results are reported in Appendix A.

### 3.1. The NAVS-G VNT

#### 3.1.1. Participants without Aphasia: HP and RHSP Groups

Verb naming test results showed a ceiling effect for each verb-type in the HP and RHSP groups (Table 3, Figure 2). For some items, synonyms or dialect words were used (the complete list is reported in the Appendix A, NAVS-G and NAT-G errors).

The logistic regression analyses did not find significant VAN (HP: *p* = 0.933, χ^2^ = 0.139, RHSP: *p* = 0.459, χ^2^ = 1.557), VAO (HP: *p* = 0.822, χ^2^ = −0.224, RHSP: *p* = 0.278, χ^2^ = 1.086) or VT fixed effects (HP: *p* = 0.925, χ^2^ = 0.893, RHSP *p* = 0.499, χ^2^ = 3.365) (see Appendix A). Notably, due to the limited variability in performance in both groups, the power to detect such effects was insufficient (range: 3–26%).

#### 3.1.2. Participants with Aphasia: LHSP Group

Only three participants with LH brain damage and aphasia showed perfect performance on the VNT (LH participants 3, 6, and 8) (see Appendix A). On average, patients with aphasia achieved 89% ± 8.18 of correct answers (see Table 3 for the mean accuracy for each verb type).

Mixed-effect regressions revealed significant effects for VAN (χ^2^ = 6.106, *p* = 0.047), and VT (χ^2^ = 17.463, *p* = 0.002), but not VAO (χ^2^ = 1.474, *p* = 0.140) (Table 4). The post hoc results, significant only for verb type, are reported in the Appendix A. In addition, verb frequency (χ^2^ = −2.774, *p* = 0.005), age (χ^2^ = −2.478, *p* = 0.013), AAT global score (χ^2^ = 2.101, *p* = 0.036), and AAT-syntax (χ^2^ = 2.555, *p* = 0.011) significantly affected accuracy on the task (Table 4).

Models for the VAN and VT were refitted by entering verb frequency, age, or AAT.sy as covariates. Notably, the AAT global score was not entered as a covariate, as scores on this measure were not obtained for all participants. Refitting the models, adding verb frequency or AAT.sy as a covariate resulted in the effect of VAN and VT no longer being significant. However, adding age as a covariate significantly improved the model as confirmed by the increase in variance explained by the model (see Table 4). The effects of age, VAN, and VT remained significant (see Table 4). The post hoc analysis of VT was not significant (Appendix A). Table 4 reports the results of the post hoc analysis of verb type. Within the obligatory verbs, the greater the number of arguments, the more difficult they were (ob3/2 < ob1 *p* = 0.023 and *p* = 0.004 respectively).

Refitting the models, adding verb frequency or AAT.sy as a covariate, resulted in the effect of argument VAN and VT no longer being significant.

#### 3.1.3. Difference between the Groups

A significant effect of group was found (*p* < 0.001, χ^2^ = 29.329, R^2^ = 0.206, power = 100%), with pairwise comparisons indicating lower accuracy for participants with aphasia compared to the other groups, and no differences between HP and RHSP (see Table 4). There was no significant interaction between the participant group and argument number (*p* = 0.13, χ^2^ = 7.099), group and argument optionality (*p* = 0.357, χ^2^ = 2.059), or group and verb type (*p* = 0.138, χ^2^ = 12.303).

### 3.2. The NAVS-G VCT

All participant groups achieved almost perfect performance on the verb comprehension test (Table 3). Therefore, logistic regression analyses were not carried out on these data.

### 3.3. The NAVS-G ASPT

#### 3.3.1. Participants without Aphasia (HP and RHSP)

Performance on the ASPT was perfect for both participant groups (Table 3).

#### 3.3.2. Participants with Aphasia (LHSP)

Only patients 1, 4, 8, 11, and 12 showed deficits whilst performing the ASPT (see Appendix A). On average, ASPT accuracy was 96.5 ± 8.32 (see also Table 3). The logistic regression analyses found a significant effect of VAN (*p* < 0.001), VAO (*p* = 0.005) and VT (*p* = 0.002) (see also Table 5 for details). Post hoc comparisons for the model with the sargument number indicated better accuracy for one-argument than for two-argument verbs (*p* = 0.01, χ^2^ = 2.936, estimates = 3.596, and SE = 1.225) and for one- vs. three-argument verbs (*p* = 0.054, χ^2^ = 1.921, estimates = 2.259, and SE = 1.176) (see Appendix A).

Clinical, but not demographic, data were significant predictors of ASPT accuracy, with accuracy being greater for participants with higher BI and lower NIHSS and mRS scores (Table 5). Clinical variables were collinear: NIHSS and mRS (R = 0.83, *p* = < 0.001), NIHSS and BI (R^2^ = −0.83, *p* = < 0.001) and mRS and BI (R^2^ = −0.88, *p* = < 0.001). Therefore, NIHSS was introduced as the only covariate in the models with VAN and VT. The main fixed effects of VAN and VT remained significant (see Table 4); post hoc comparisons indicated better accuracy for one-argument verbs than for both obligatory and optional three-argument verbs.

#### 3.3.3. Difference between the Groups

There were no significant differences between groups, not even in interaction with other factors (argument number, argument optionality, verb type).

### 3.4. The NAVS-G SPPT

#### 3.4.1. Participants without Aphasia (HP and RHSP Groups)

Table 3 reports SPPT mean accuracy in HP and RHSP groups. In the HP group, logistic regression analyses found a significant effect of canonicity and of sentence type (Table 5, Figure 3). However, the effect of canonicity disappeared after removing relative sentences. The post hoc results are reported in Appendix A.

Accuracy on the SPPT was better for individuals with higher education (qualification and graduation), and for younger vs. older participants (see Table 6, Figure 4). The models for canonicity and sentence type were refitted, adding education qualification as a covariate, as this was the most predictive factor. Both main effects remained significant (see Table 6). Post hoc comparisons for the model sentence type and education indicated that accuracy on OR sentences was lower than for all other sentences, including passives, SR and OWQ (all *p* < 0.001).

In the RHSP group, logistic regression analyses showed a significant effect only for sentence type, age and Ed.q. (Table 6). The post hoc results are reported in Appendix A. The model for sentence type was refitted by adding Ed.q. as a covariate; this analysis showed a significant main effect and post hoc comparisons indicated lower accuracy for relative than non-relative sentences, and for OR than non-OR and P sentences (see Table 6, Appendix A). Error analyses showed that participants produced passive, subordinate clauses (53%), or SR (47%) rather than OR sentences.

#### 3.4.2. Participants with Aphasia (LHSP)

All LHSP group participants showed difficulty in sentence production on the SPPT (Table 3 and Appendix A). Moreover, for all patients, (1) canonical sentences were produced better than non-canonical sentences, (2) OR accuracy was lower than for other sentence types, and (3) each study participant demonstrated a deficit in OR sentences and (other than HP) combined OR impairments with deficits in another NAVS-G or NAT-G subtests, mostly the VNT. The three patients without VNT impairments presented deficits in OR and in SR, LHSP8 and in ASPT, and LHSP3 in NAT_OWQ.

Table 6 and Figure 3 summarize the results of the logistic regression analyses for the SPPT. Results showed significant effects of canonicity, even after removing relative sentences, and of sentence type (post hoc results were reported in Appendix A). No significant effects were found for age and education (Figure 4)*;* whereas accuracy on the SPPT was better for participants with lower NIHSS scores and higher AAT.ss, AAT.sy, and aphasia severity scores.

Most of these variables are correlated with each other: NIHSS with AAT.ss (R^2^ = −0.725, *p* = 0.002); NIHSS with AAT.sy (R^2^ = −0.624, *p* = 0.0127); NIHSS with aphasia severity (R^2^ = −0.713, *p* = 0.006); AAT.ss with AAT.sy (R^2^ = 0.821, *p* < 0.0001); AAT.ss with aphasia severity (R^2^ = 0.711, *p* = 0.006); and AAT.sy with aphasia severity (R^2^ = 0.667, *p* = 0.012). Given the collinearity between clinical variables, models for canonicity and sentence type were refitted using the most predictive variable (AAT.ss) as a covariate (Table 6). Analyses showed that both canonicity and sentence type remained significant. Post hoc comparisons indicated better accuracy for active than passive, SR and SWQ sentences; better accuracy was also found for SWQ than for SR and OWQ. In addition, the accuracy for OR sentences was lower than for passive, SR and OWQ sentences. These results were the same when the covariate was not included in the analysis, except that the R^2^ increased from 0.19 to 0.61. Models for canonicity and sentence type were also refitted with NIHSS as a covariate (Table 6). The main effects of canonicity and sentence type remained significant, and so did the post hoc comparisons.

#### 3.4.3. Difference between the Groups

Table 6 shows that performance of patients of the LHSP group on the SPPT was worse than both HP and RHSP group participants. No significant difference was found between HP and RHSP (Appendix A).

The interaction between group and canonicity was not significant; however, the interaction became significant after excluding relatives (Table 6). A significant interaction was also found between group and sentence type; post hoc comparisons revealed that LHSP participants were significantly impaired, compared to HP, on all sentence types except actives and passives, and that their accuracy was lower than that of RHSP on SR and OR (Table 6).

### 3.5. The NAVS-G SCT

#### 3.5.1. Participants without Aphasia (HP and RHSP Groups)

Logistic regression analyses did not find significant effects for sentence type and canonicity for either the HP or RSHSP groups (see also Appendix A).

#### 3.5.2. Participants with Aphasia (LHSP Group)

There was a significant main effect of canonicity (see Table 7). This effect remained significant even after excluding relative clauses (see Table 7). The post hoc results are reported in Appendix A.

Accuracy on the SCT was better for participants with lower NIHSS scores and higher scores on measures of AAT.ss, AAT.sy, BI, and aphasia severity (Table 7). Given the collinearity between clinical variables, models for canonicity and sentence type were refitted using the most predictive variable, AAT.ss, as the covariate (Table 7). Analyses showed that both canonicity and sentence type remained significant. Post hoc comparisons showed that relative sentences, and specifically OR sentences, were more difficult to understand than all other structures.

#### 3.5.3. Difference between the Groups

The aphasic group performed worse than both HP and RHSP groups (see Table 7). The interactions between group and sentence type, or between group and canonicity, were not significant (Table 7).

### 3.6. The NAT-G

#### 3.6.1. Participants without Aphasia (HP and RHSP Groups)

The healthy participants’ performance was 100% accurate across all sentence types (Table 3). Three participants in the RHSP group had deficits on the NAT (see Table 3 and Appendix A). Logistic regression analyses found no significant effects of canonicity or sentence type (see Table 7 and Appendix A for details).

#### 3.6.2. Participants with Aphasia (LHSP Group)

Two patients (13 and 14) did not want to perform the NAT. Patients 4, 5, 6, and 15 achieved perfect NAT performance (see Appendix A).

Logistic regression analyses found significant effects of canonicity (Table 8). Accuracy was better for canonical than for non-canonical sentences, even when excluding relative sentences. Sentence type was also a significant predictor of accuracy (all *p* < 0.001). (Table 8). Results of the post hoc tests are reported in Appendix A. Accuracy on the NAT was also higher for participants with higher scores on the AAT syntax and AAT spontaneous speech, as well as on the Token Test and aphasia severity scores, and for participants with lower NIHSS scores (see Table 8).

Because most of these variables were collinear, we refitted models for canonicity and sentence type by adding the strongest predictor of accuracy among them, i.e., the AAT.ss When refitting the model for sentence type with this covariate, the main effect remained significant (*p* < 0.001), with the R^2^ increasing to 0.62 (Table 8). Post hoc comparisons showed that accuracy was lower for passive than active sentences (*p* < 0.001). In addition, SR sentences were more difficult than active sentences (*p* = 0.005) and SWQ (*p* < 0.001), and accuracy on OWQ was lower than SWQ (*p* = 0.005). OR were the most difficult sentences (OR > non-OR: *p* < 0.001, OR > *p*: *p* = 0.005, and OR > OW: *p* = 0.002), even if there was not a significant difference between OR and SR (see Table 8). When refitting the models with the NIHSS score as a covariate, canonicity and sentence type both remained significant, and so did all the above post hoc comparisons (Table 8).

For the NAT, the error analysis was difficult. However, typically, patients were not able to order the word cards and did not complete the tasks.

#### 3.6.3. Difference between the Groups

Accuracy on the NAT-G was significantly different among groups (*p* = 0.001, χ^2^ = 13.309, R^2^ = 0.2. power = 88%): patients with aphasia (LHSP) were worse than HP (*p* < 0.001, χ^2^ = −0.861) and RHSP (*p* = 0.036, χ^2^ = −2.262). No significant difference was found between HP and RHSP (*p* = 0.162, χ^2^ = 1.398).

The interaction between group and canonicity was significant after removing the relative sentences (*p* = 0.018, Table 8), and post hoc comparisons indicated that patients with aphasia had difficulty in performing non-canonical sentences—passive and OW—compared to HP (χ^2^ = −3.774, *p* < 0.001) and RHSP (χ^2^ = −2.25, *p* = 0.037).

Although the interaction between group and sentence type was marginally significant (*p* = 0.053, χ^2^ = 18.143, R^2^ = 0.20), the effect of group was not significant for any of the sentence types. This is likely due to the limited number of participants for each level of the variables (as also indicated by power ranging from 0 to 13%).

### 3.7. NAVS-G and NAT-G Errors’ Analysis in Participants with Aphasia (LHSP Group)

Table 9 and Appendix A report the error analysis. In VNT, the LHSP group most frequently produced semantic paraphasias. For one- or two-argument verbs, errors consisted of substituting the target verb with a verb with the same argument number, (e.g., to push “schieben” instead of to pull “ziehen” or to kick “strampeln” instead of to crawl “krabbeln”). For three-argument verbs, target verbs were substituted for another verb with a lower number of arguments (e.g., to get “holen” instead of to give “geben”). Eighteen percent of errors consisted of verb omission, with better performance on ob2/3 verbs compared to op2/3 verbs (op2 versus ob2/3, *p* = 0.28 and *p* = 0.004, respectively; and op3 versus ob3 *p* = 0.02) (see Figure 2).

In ASPT, missing verb conjugations and positions, argument omission, and incorrect ordering of arguments within the sentence were the most frequent errors.

At the sentence level, the most frequent errors were the incorrect word order, the production and comprehension of canonical instead of non-canonical sentences, and the production of passive, subordinate sentences instead of OR sentences.

### 3.8. Correlation and Comparison across the Subsets

Pearson-correlation analyses were conducted across all subtests of the NAVS-G and NAT-G (see Table 10), showing a significant correlation between each subtest and the global NAVS-G and NAT-G scores, between VNT, ASPT, SCT and between SCT, NAT-G, and SPPT.

The results of the comparison between the subtests using Wilcoxon tests are reported in the Table 11. The performance on the VNT was worse than on ASPT (*p* = 0.003) and VCT (*p* = 0.002), but it was better than on the SPPT (*p* = 0.003). Accuracy on the NAT was significantly better than in SPPT (Z = −3.853, *p* < 0.001). Even the post hoc comparison of OR clauses showed that NAT was better performed than SPPT (Z = −2.392, *p* = 0.017). SCT was better performed than the SPPT (Z = −3.413, *p* = 0.001).

## 4. Discussion

This study presents the adaptation of the NAVS and NAT to the German language, documenting its ability to detect grammar deficits in (German) stroke patients with chronic mild to minimal language impairments. Moreover, the present research demonstrates that nonlinguistic factors, such as age, education and stroke-related factors (such as stroke severity and stroke localization), contribute to variability in test performance in different ways, in participants with and without aphasia.

### 4.1. The Adaptation to the German Language

The German-language adaptation of these tests was successful, as 27 HP exhibited high agreement in both verb and sentence production tasks (see Table 3 and Appendix A).

The validity of the original NAVS is supported by standardization procedures and psychometric data for test validity and reliability that are provided in the respective manual. Several publications have reported the results of testing different populations and languages with the NAVS and NAT: English [8,102,103], Italian [7], and Chinese [36]. All studies have shown that both NAVS and NAT can detect and measure syntactic deficits in stroke patients and patients with primary progressive aphasia. This study provides a replication of the results of the original version in participants with mild to minimal language impairments and in individuals without aphasia. Notably, the study did not aim to provide a standardization of the NAVS-G and NAT-G for the German population, as the latter would require a larger number of participants and a greater range of severity of language deficits than reported in this study.

#### 4.1.1. NAVS-G for Testing Syntactic Competence at the Verb Level

In line with previous versions of the VNT [7,8,103], the VNT-G identified deficits in verb production, which were specific to individuals with aphasia and increased with the complexity of the verb argument structure (Table 4 and Appendix A). These results confirmed that these deficits are not limited to patients with Broca’s aphasia and concomitant agrammatism [85,86,87,89], but rather demonstrated for the first time (to the best of our knowledge) that the number and type of verb arguments affected naming accuracy, even in patients with residual aphasia.

Like in Barbieri et al. [7], these two factors did not affect verb naming accuracy in healthy participants (Table 4), who scored 99.6 to 100% accuracy on the VNT-G subtests (Table 3). Similarly, individuals who suffered a right-hemisphere stroke and had no aphasia (RHSP) did not exhibit any difficulty on the VNT-G. Conversely, VNT-G performance was worse in patients with aphasia than in participants without aphasia, including the RHSP group (Table 4). This finding supports the idea that left, but not right, hemisphere lesions lead to syntactic impairments at the verb level. In line with this, several neuroimaging studies have shown greater recruitment of neural processing resources in the left (vs. right) hemisphere for verb naming with increasing arguments [31,108,155,156,157].

Our results showed that, in participants with aphasia, age affected verb naming (Table 4, see also Section 4.2.1 for discussion); nevertheless, the effect of verb argument structure complexity was not accounted for by age.

More importantly, our results indicated that verb-naming deficits in aphasia are not a byproduct of lower verb frequency. According to Hebbian learning [158,159,160,161], it is expected that the higher the verb frequency, the easier the verb is to recall. Logistic regression identified a counterintuitive effect of verb frequency on accuracy on the VNT, i.e., verbs with the highest frequency were associated with the lowest naming accuracy (Table 4). In this regard, however, it is interesting to note that in English, frequent verbs have a greater number of arguments than less frequent verbs [162]. Within this perspective, a verb’s frequency could inversely relate to syntactic complexity, i.e., the poor performance on highly frequent verbs on the NAVS-G VNT could mask the effect of argument structure complexity. In favor of this interpretation, differences in accuracy based on verb type and argument number disappeared after introducing frequency as a variable in the regression model. Moreover, previous studies found that verb frequency, when matched across verbs with different arguments, does not affect verb retrieval [8,108]. However, frequency does affect verb production accuracy when verb argument is not controlled [163,164], presumably due to its facilitating role in lemma access [159,165]. More studies, mainly linguistic studies, should be conducted to support or replicate this novel finding.

The error analysis in patients with aphasia supports the idea that verbs are lexical items, which possess syntactic argument valence [6,7,8,9,10,11]. Patients’ low accuracy on the VNT, the significant effect of argument structure complexity (in particular within the obligatory verbs, Table 4), and the finding of frequent omission errors (Table 9), in conjunction with the near to perfect performance in the comprehension of the same verbs (see also Table 11: the difference between VNT and VCT was significant) point to a deficit in accessing verb representations and not to an impairment of the stored verb representation. Remarkably, the error analysis revealed that patients often substituted the target verb with a verb that had a similar meaning but fewer arguments or nouns, and used infinitive verbs by producing active sentences in ASPT (Table 9). Thus, we interpreted that verb meaning was available to our patients, but their hierarchical valence was not. Patients also showed semantic paraphasias among verbs with the same number of arguments (Table 9). This finding suggests that access to verbs, based on argument structure, was available, albeit the selection of these verbs was impaired. These two kinds of errors—substitution of verbs with lower argument valence and semantic paraphasias—suggest that the verbs are stored lexically (i.e., as a static representation) and hierarchically, based on argument structure properties.

Our results corroborated the previous finding that the obligatory versus the optional status of verb arguments may also contribute to verb deficits present in aphasia (Table 5) [8,166]. Previous data, however, have reported contrasting findings. In the original NAVS, the optional verbs were more complicated than obligatory for both verb naming and sentence production [10]; the opposite was found for the NAVS-I [7]. Our results were, to some extent, more complicated: the effect of optionality was not significant and there was no difference between optional verbs (independently of verb arguments) and obligatory one-argument verbs. However, optional two- and three-argument verbs were better named than their obligatory counterparts. These data might be partially explained by the saturation theory [108,167,168,169]. This theory assumes that only one complementation frame is represented in the lexical entry of verbs with optional complements, and when such a verb is inserted into a sentence without a complement, a particular operation (saturation) is executed to take care of the unassigned thematic role and, consequently, allows the omission of the complement [167,168]. In this framework, we expected, however, that op3 verbs would be better processed than ob2. In our study, accuracy in op3 verbs (such as “schreiben” to write), which takes a minimum of two arguments, did not differ from ob2 verbs. Therefore, the saturation theory cannot fully explain our results.

Our data support the idea that testing verb argument processing in patients with aphasia is crucial, as this impairment may also affect the production of active sentences [166]. Within the NAVS-G, the ASPT mirrored the patterns found on the VNT, albeit to a lesser extent. The difference between the control and the aphasic groups was not significant. This finding was not surprising, as agrammatic sentence production often consists of simple, canonical, SV, and SVO structures compared to the ones examined in the original ASPT [8]. Moreover, the NAVS-G ASPT, as opposed to the NAVS-G VNT, did not require verb retrieval, as the verb and its arguments are provided. Instead, this test asked participants to build a sentence by mapping the verb arguments onto the sentence structure. On this test, aphasic participants showed difficulties when the number of verb arguments increased (Table 5 and Table 9), suggesting a reduced ability to build the syntactic and thematic structures on which a sentence is constructed. In our study, these deficits were selective for patients with aphasia, as participants in the control groups did not make any errors in ASPT (Table 3).

It is interesting to note that only five patients demonstrated ASPT deficits, but among these, two suffered from residual aphasia. Thus, the recognition of such deficits at the verb-argument level in patients with residual aphasia could be crucial, as identifying specific impairments will enable a more focused therapeutic approach [135,170].

#### 4.1.2. NAVS-G and NAT-G for Testing Syntactic Competence at the Sentence Level

As for verb processing, the control group exhibited ceiling accuracy on sentence-level processing on both the NAVS-G and NAT-G, excluding relative sentences and, to a larger degree, object-relative (OR) structures (Table 3, Table 6 and Table 8). The finding that the production of OR clauses exerts high processing demands even in healthy participants is in line with previous studies (e.g., [171,172,173]) across several languages, including English [174,175], German [176], and Spanish [177]. Electrophysiological studies in young, healthy participants showed a robust N400-P600 complex during the processing of OR clauses, a pattern that is functionally linked to syntactic reanalysis or ambiguity resolution [178,179]. Our data also confirmed the widely acknowledged performance asymmetry between OR and SR (see Table 6 and Table 8; Figure 3) [39,53,96,180]. Thus, the poor OR accuracy on the NAVS-G SPPT in both groups of participants without aphasia documented an additional example of its sensitivity towards identifying critical aspects of syntactic processing within the German language.

In patients suffering from aphasia, all subtests involving sentence processing—NAVS-G SPPT and SCT and NAT-G—showed that sentence-type and canonicity (i.e., canonical vs. non-canonical word order) affected both sentence comprehension and production ability (Table 6, Table 7 and Table 8). This finding is in line not only with the results of the preceding version of the NAVS and NAT [7,8,36,102], but also with a large number of previous studies (production: [39,53,96,180]; comprehension: [35,53,92]). Consistent with Schröder et al. [71], Caramazza [93], Nespoulous et al. [181], and Cho-Reyes and Thompson [8], sentence comprehension was better preserved than production (Table 11). In addition, both tests revealed syntactic deficits in patients with residual aphasia. Together with evidence of errors on the SPPT, which most frequently consisted of production of sentences with incorrect word order (Table 9, Appendix A), these results suggest that difficulties in processing syntactic movement could be encountered by patients with residual aphasia.

These results support the use of the NAVS-G and NAT-G for detecting syntactic deficits in patients with aphasia. Because the outcomes of syntactic therapy are generally modality-specific [66,96,97], such data are informative for developing effective treatment for German-speaking people with aphasia.

In keeping with previous studies [8,75,76,181,182,183], we also found that the production of non-canonical sentences was more impaired than the production of canonical sentences for patients with aphasia, and this was independent of aphasia type and severity (Table 6 and Table 8). This effect was not exclusive for patients with aphasia, as participants without language impairments also yielded more difficulties in processing OR than SR. However, in patients with aphasia, this difficulty was significant for all sentence types (the canonicity effect remained significant after excluding relative clauses). This study documented that difficulty with non-canonical sentence production was present, not only in agrammatic individuals [181,182,183], but also in patients with minimal language impairments. Moreover, both NAVS-G SPPT and NAT-G allowed a detailed differentiation within non-canonical sentences by allowing a comparison between sentences entailing NP-movement (i.e., passives) and sentences entailing Wh-movement (i.e., object Wh-questions and object relatives). Differences in accuracy between object-relative and passive sentences reported on the SPPT-G (Table 6) and NAT-G (Table 8) corroborated the evidence from the previous literature [74,184,185,186,187], which points to qualitatively different processing underlying sentences containing Wh- vs. NP-movement. It has been discussed that OR sentences are more syntactically complex than passive sentences [74,185], as they differ in their underlying syntactic movement, entail sentence embedding, and include a greater number of nodes in the syntactic tree [1,4,22]. Adults in this study—like older children [188]—from both the experimental and control groups, resorted to passive constructions when the production of ORs was elicited. Treatment studies suggest that the type of syntactic movement (i.e., NP- vs. Wh-movement) is an essential factor to consider for aphasic individuals with sentence deficits, as training sentences with Wh-movement does not affect production or comprehension of sentences with NP-movement (e.g., object relative structures to object Wh-question forms) [7,67,69,72,73,74,75].

Moreover, patients with aphasia also showed significant impairment in production of canonical sentences (in ASPT and SPPT) ([99,100]). Such deficits could not be detected using existing German syntax tests [81,82,83]. Better specification of the grammatical impairment in patients with aphasia, particularly in the presence of minimal aphasia, is crucial for devising targeted and effective interventions that may help recovery of language functions.

Notably, similar performance patterns were shown in sentence production as tested by the NAVS-G SPPT and NAT-G, where scores for the two measures were highly correlated (Table 10). However, higher accuracy was found on the NAT compared to the SPPT-G (Table 11), suggesting that although both tests measure the same syntactic competence, the SPPT-G is more sensitive to this aspect. The NAT-G minimizes the impact of word-finding deficits, motor speech impairments, buccofacial apraxia, and working memory abilities on performance [23,103] in that word cards are used to facilitate lexical access and the nature of the test (i.e., anagram assembly: arranging word cards) allows patients to use nonlinguistic processes (e.g., offline trial and error rearrangement of the cards) to complete the task. In contrast, the NAVS-G SPPT requires online production of target sentences. That said, the NAT-G is relatively brief and easily administered and could be a clinically convenient measure of syntactic processing.

#### 4.1.3. Sensitivity in Detecting Aphasia

The comparison between the aphasic group and the control group indicates that the combination of the NAVS-G and NAT-G was able to identify language impairments in a target population, namely, stroke patients with aphasia. Patients with aphasia, compared to the control group without any history of aphasia, performed significantly worse in all subtests, except VCT and ASPT (Table 4, Table 5, Table 6, Table 7 and Table 8). Therefore, the distinction between the target group and the control groups encompassed both word- (verb) and sentence-level processing, and both comprehension and production, i.e., it was clear in each leading aspect of the language faculty [3,4,5,189].

On tests of verb argument processing, the aphasia group performed relatively well (VNT mean accuracy was 89.4 and ASPT reached 96.5, see Table 3), and the group effect was significant but not in interaction with verb type or verb argument (Table 4 and Table 5). Differences in performance between aphasic and non-aphasic groups were most evident in sentence-level processing, where accuracy differed between groups on all the subtests (NAVS-G SPPT and SCT and NAT-G and, see Table 6, Table 7 and Table 8). Remarkably, these differences were significant in comparison to both HP and RHSP. On the SPPT, the most demanding test for all participants, the difference between participants with and without aphasia was shown for both canonical (SR) and non-canonical (OR) sentences. On the SPPT, accuracy for RHSP was lower than for HP (at least for OWQ). However, this population showed better production of complex sentences than patients with the same demographic characteristic, stroke clinical severity, and infarct localization, but on the left hemisphere (Table 6, Table 7 and Table 8). These results were expected, as functional imaging studies on sentence processing have shown that regions homologous to language-related brain regions in the right hemisphere were active during syntactic processing, but the left hemispheric dominance remained [44,190,191].

Adjusting the regression analyses with the scores of the AAT syntax, the effects of verb-argument structure complexity were no longer significant. As these scores reflect the syntactic structure of spontaneous speech production [141], by quantifying the completeness and complexity of sentence patterns or of the correct inflections, syntactic aspects of functional communication and processes underlying NAVS-G and NAT-G performance might largely overlap. Conversely, after adjusting the regression models with the AAT spontaneous speech score, the effect of sentence-type and syntactic complexity (i.e., canonicity) remained significant, suggesting that production aspects that are taken into account by the AAT spontaneous speech (i.e., articulation and prosody, semantic and phonological speech structure) could only partially explain the low sentence production in patients with aphasia.

The significant relationship between performance on several language scores (AAT global score, TT, and aphasia severity) (Table 4, Table 5, Table 6, Table 7 and Table 8) and on the NAVS-G and NAT-G is not surprising since grammar is an essential part of the language faculty [2,3,5]. Therefore, if grammar is impaired, we expect a simultaneous disorder of language. In line with this statement, grammatical impairments have been well documented in participants with aphasia, and are not limited to agrammatism [34,85,86,87,89].

In this study, the NAVS-G and NAT-G detected grammatical impairments in patients with minimal language deficits. The possibility that such patients may suffer from grammatical impairments has clinical implications on both assessment and intervention for people with aphasia. First, the current best method to diagnose residual aphasia is the functional communication analysis [136]. This method is time consuming and requires highly trained speech therapists, two factors that complicate the diagnosis of residual aphasia. In this respect, the use of NAVS-G and NAT-G could provide a time- and cost-efficient method to quantify (even mild) grammatical deficits in residual aphasia, thereby avoiding reliance on a functional communication analysis. On the other hand, language impairments constitute a daily burden for the people affected by them, and even for people with residual aphasia, as they limit participation and professional reintegration [136,170]. Moreover, therapy is not very effective in this patient group, as shown by the fact that minimal residual language impairments remained stable even for years post-stroke, in at least 20% of participants with residual aphasia [136,170]. Therefore, a better characterization of language impairments in these patients could help target language interventions, thereby improving their efficacy. In this respect, the NAVS-G and NAT-G aphasia battery could be a useful tool for identifying syntactic deficits in this patient group. To verify this finding, the replication of the present results in a larger sample of participants is necessary. At this stage of the study, it was not our intention to provide a validation and standardization of the NAVS-G and NAT-G. Nevertheless, we do not expect these tests to be specific for the stroke aphasic patient even in the future validation phase. Rather, we expect that (and will test for) these tests will prove useful for detecting syntactic deficits in other clinical populations, such as individuals with neurodegenerative dementia due to Alzheimer’s disease and related disorders, or children with language impairments [34].

### 4.2. The Importance of the Covariates’ Control

One of the aims of the study was to achieve bias reduction. Therefore, not only we well-matched participants in the three comparison groups of the experimental conditions, but also controlled the effects of several variables that are known to affect (language) task performance (although they are not known to be directly related to syntax) [107,192]; namely, it has been shown that demographic information, lesion size, and atlas-based lesion location predictors accounted for almost 60% of the variance in speech production [118]. Our data could not confirm that these factors were essential for explaining grammar defects in aphasia, but some of them were relevant for the control groups.

#### 4.2.1. The Covariate “Age”

Our results found that the participants’ age might explain sentence production difficulties (Table 6, Figure 4), as younger participants showed better accuracy than elderly participants. This effect was significant for HP and RHSP, but not in patients with aphasia. In this group, the participants’ age explained, to some extent, the variance on the verb naming task (Table 4).

It is well-known that age is a very crucial factor, which influences acquisition of complex grammatical structures in children [193]. Recent functional imaging studies attribute this finding to the maturity of structural and functional brain connectivity within the language network [194,195,196,197,198]. Similarly, it has been shown that significant changes in functional and structural brain connectivity can be found in aging, and such changes may explain the cognitive decline in elderly people [199]. However, while working memory [200] and word production are generally acknowledged to be compromised in healthy elderly adults [201], there is no general consensus about the presence of actual syntactic deficits in this group [116,201,202,203].

Our data found that age affected overall accuracy on the NAVS-G VNT and SPPT, but the effects reflecting syntactic complexity remained significant (Table 4 and Table 5). Taken together, these findings suggest that aging may not specifically impair syntactic competence (i.e., processing of verb argument structure, word order, or syntactic complexity). Age-related resource-based limitations, rather, might be associated with reduced syntactic memory span [204]. However, our data could not confirm this interpretation as age did not specifically affect accuracy of OR sentences (Table 6), which are the most demanding in terms of working memory resources than other sentence types [23,24,25]. Instead, we propose that aging could be linked to a more general reduction in efficiency with regard to integrating various neuronal and cognitive resources, which, in turn, affects accuracy in oral sentence production. More recently, it has been shown that despite age-related declines in the gray matter of language-related brain regions [203,205,206], the functionality of the language network does not show age-related differences in within-network connectivity or responsiveness to syntactic processing demands [202,207,208]. It is the connectivity between the language and the default mode network that decreases with age [202,207,208]. Namely, older adults often show difficulties in deactivating the default mode network during task performance [208], which leads to a more diffuse and less focal recruitment of task-relevant brain networks (e.g., [207]). This finding suggests that, even if a network remains functionally intact with age, its ability to interact with other networks in the service of task goals may be affected. This hypothesis might explain why NAVS-G SPPT was performed more poorly than NAT-G (Table 3 and Table 11), given that oral sentence production requires integration of several cognitive competencies, including lexical retrieval, phrase structure building, thematic mapping, and executive control during sentence planning [25,176,177,209]. Conversely, the NAT-G, in which participants are provided with the lexical items that constitute the target sentence, requires fewer processing resources than the SPPT-G.

The finding that syntactic competencies could relate to aging needs to be replicated in a large sample (and ideally, evaluated together with functional imaging measures such as resting-state fMRI or DTI); nonetheless, it remains an essential finding in this study. Only a few language tests (the Token Test for example) take “age” into account for the validation process, i.e., introducing correction scores for this confounding parameter. Normative scores will allow us to distinguish aging sentence error from aphasia symptoms. Due to the increasing aging of the German population, the implications of these findings are relevant.

#### 4.2.2. The Covariate “Education”

This study found that both control groups performed worse on the OR than the other sentences of SPPT, and this test performance was significantly related to the participants’ educational qualification (Table 6). Moreover, in the interaction between group population, sentence type, and education, the post hoc analysis found a significant effect for the control groups (Table 6, Figure 4) (and specifically on OR, even if at an uncorrected level, see Appendix A). Namely, accuracy was reduced in participants with lower education, but only for healthy participants and right-hemisphere stroke patients.

Education’s influence on language performance has been repeatedly shown [210,211,212] even for grammatical processing [213]. However, to the best of our knowledge, this is the first study postulating that education may explain the variability in OR sentence production accuracy in healthy participants (Figure 3). The analysis of participant error production during SPPT-G showed that control participants, instead of producing OR clauses, produced passive, subordinate clauses or SR, which require fewer working memory demands. Since educational achievement is known to correlate with working memory capacity [213], differences in working memory could, in principle, be responsible for the education-related differences observed in the OR task. However, Chipere [214] showed that, after memory training, less educated participants improved their performance in a memory task, but not in a sentence-comprehension task. Therefore, Chipere [214] concluded that the poor performance of the less educated speakers on a comprehension task should be attributable to a lack of linguistic knowledge and is not limited to working memory capacities.

Compared to other sentence types, relative sentences, and particularly OR, are used less frequently in functional communication [215]. Relative sentences, compared to other sentence types, are syntactically more complex [3,4,5] and have different communicative functions [215,216]. The idea of a “syntactic” frequency hierarchy is increasing in popularity [217,218,219,220]. Reali and Christiansen [221] showed that the frequency of co-occurrence of the word combinations (or chunks) forming the OR clause influences processing difficulties in English. Participants judged OR in the high-frequency condition—O (VS) word order within the clause—as easier than their O (SV) counterpart in the low-frequency processing of nested syntactic structure. Street and Dabrowska documented that education influences the ability to produce nouns with a correct inflection [222,223], as well as the ability to process passive [224] and implausible sentences [225], and attributed this effect to the asymmetries in distributing these syntactic constructions in spoken and written discourse. Their data provide evidence that difficulties in less educated speakers lead to reduced exposure to specific linguistic constructions and, thus, results in the absence of well-entrenched syntactic schemas. Performance of these speakers improved dramatically after additional experience with the relevant construction, showing that the initial differences in test scores are attributable to differences in familiarity with specific linguistic constructions [117].

Although we did not find data specifically linking relative and OR processing, frequency of its grammatical construction, and education—since even in German, more educated speakers (i.e., with the highest educational degree) have more experience with formal written language—we speculate that education might influence the most demanding operations, such as OR clauses, which occur less frequently. The “syntactic’’ frequency may anchor these sentences’ mental representations, somewhat like a chunk [226]. Due to the increased practice of literacy skills, syntactic constructions, which are commonly available, might be better “trained”, perhaps in an analogous way, as speech therapy can induce an improvement of impaired OR clause processing in a patient with aphasia.

Future studies will need to replicate these findings to better differentiate the effect of education from the effect of syntactic complexity. Nevertheless, it is remarkable that after refitting the logistic regression model and adding the covariate education (Table 6), the greater difficulty in producing OR (vs. other) sentences remained significant.

#### 4.2.3. The Role of the Right Hemisphere

Our results on right hemisphere patients were, to some extent, unexpected. A specific comprehension impairment, such as difficulty in processing syntactic markers, has been previously described [115,227]. However, our patients showed perfect performance on measures of verb and sentence comprehension (NAVS-G VCT and SCT). Conversely, our right-hemispheric brain-damaged patients demonstrated deficits in sentence production on both the SPPT-G and NAT-G when syntactic complexity increased [42,179,228]. The effect of canonicity was not significant in this group, and they showed the most difficulties in the production of OR, SR, and OWQ sentences (Table 6 and Table 8), with poorer performance on OWQ than non-brain-damaged (HP) participants. However, this effect was not significant when comparing patients with aphasia (Table 6 and Table 8). It is remarkable that clinical scores were not predictors. i.e., stroke severity cannot explain their production disorders. Even age and education only partially explained a sentence type effect, which, despite having only 15 participants, was significant and had sufficient power and effect size (Table 6). In accordance with studies discussing the crucial role of this hemisphere in thetical grammar, we postulate that integrating parts of a text into a coherent whole is relevant for relative sentence and objective question processing [229]. These findings were novel and were limited to 15 patients. More research is necessary to replicate the current findings, and to better understand the role of the right hemisphere in syntax processing, which remains unclear.

#### 4.2.4. The Covariates “Clinical Scores”

Results of the regression analyses indicate that NAVS-G and NAT-G performance was affected by stroke severity. For the ASPT, all measures of stroke severity (BI, mRS, and NIHSS) affected task performance. Since the model’s power and effect size improved when refitting the models with these variables, stroke severity partially explained accuracy on the ASPT. Nevertheless, the difference between verbs with three valences and one valence remained significant even when stroke severity was taken into account (Table 5).

The NIHSS score predicted performance on all NAVS-G and NAT-G subsets (Table 5, Table 6, Table 7 and Table 8), as did aphasia severity and spontaneous speech competence. It is well-known that NIHSS reflects clinical severity as well as the severity of aphasia, and it has been shown to correlate with the prognosis for stroke-induced aphasia [230,231,232]. Surprisingly, we did not find a relation between NIHSS and NAVS-G and NAT-G performance in the RHSP group. The presence of concomitant neurological symptoms reduced the ability to correctly produce sentences with diverse complexity [233,234,235]. Although these findings need to be replicated in a larger sample of subjects, they provide evidence that selective deficits in grammar are affected by factors that lie outside of the “language proper” [34].

However, despite the significant effect of NIHSS for explaining the variance in grammatical test performance, the effect of sentence type and canonicity remained, providing evidence that more specific syntactic factors (i.e., syntactic complexity) influenced sentence production in aphasia. In line with this interpretation, the RHSP group, which did not differ from the patients with aphasia for NIHSS, BI and mRS, performed significantly better on NAVS-G and NAT-G (Table 6, Table 7 and Table 8).

## 5. Conclusions

The present study found that the combination of NAVS-G and NAT-G was able to identify grammatical deficits in native German-speaking stroke patients, even in those with mild, residual language disturbances. This study, therefore, provides the foundation for developing a novel and flexible test battery for testing grammar in aphasia.

## Figures and Tables

**Figure 1 brainsci-11-00474-f001:**
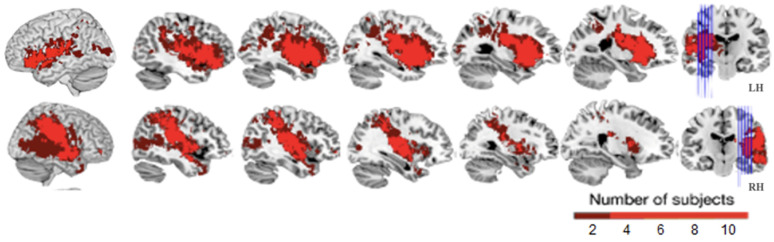
Overlap of the binarized stroke lesions in the left hemisphere (LH) and right hemisphere (RH) in patients with aphasia and without aphasia respectively. The color bar indicates the degree of overlap of lesions: the darker the color, the fewer the patients with a lesion in this area.

**Figure 2 brainsci-11-00474-f002:**
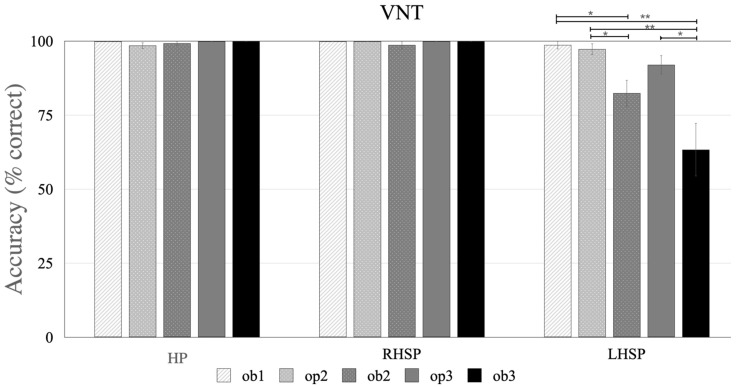
NAVS-G VNT results: The bar charts display the results in the healthy participants (HP), the right hemispheric stroke patients (RHSP) without aphasia, and the left hemispheric stroke patients (LHSP) with aphasia for verb naming with one argument (ob1), optional two and three arguments (op2/3), and obligatory two and three arguments (ob2/3). Participants without aphasia show a ceiling effect across verb-types. LHSP patients display worse performance for ob 2/3 versus op 2/3 verbs and, in obligatory with more than one argument (* *p* < 0.05, ** *p* < 0.005).

**Figure 3 brainsci-11-00474-f003:**
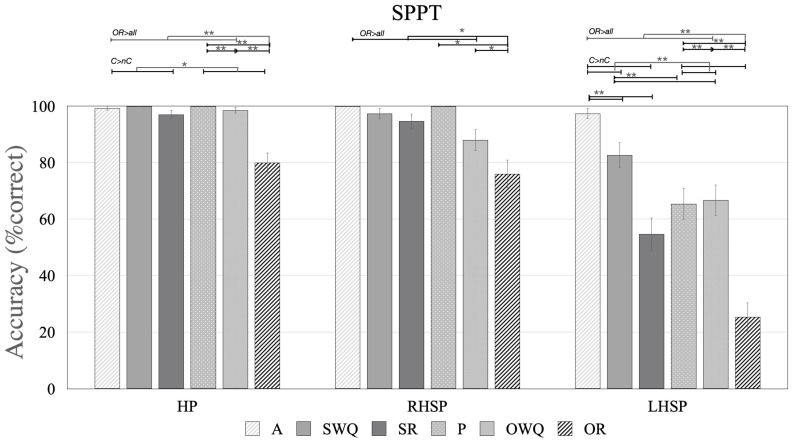
NAVS-G SPPT result: The bar charts display results in the healthy participants (HP), the right hemispheric stroke patients (RHSP) without aphasia, and the left hemispheric stroke patients (LHSP) with aphasia for production of several types of sentences: A = active, P = passive, SWQ = subject Wh-question; OWQ: object Wh-questions; SR = subject relative; OR = object relative. Full color bars indicate canonical sentences, and crackled color bars indicate non-canonical sentences. In HP and LHSP groups, non-canonical sentences (nC) are performed more poorly than canonical (C) sentences, but in LHSP the effect remained significant after excluding relative sentences. In all three groups, relative (Rel) accuracy is worse than non-relative (nonRel) sentences and OR shows the worst accuracy; OR sentences show the worst accuracy in the LHSP group. In these patients, accuracy decreases with increase of syntactic complexity also within canonical sentences. * *p* < 0.05, ** *p* < 0.005.

**Figure 4 brainsci-11-00474-f004:**
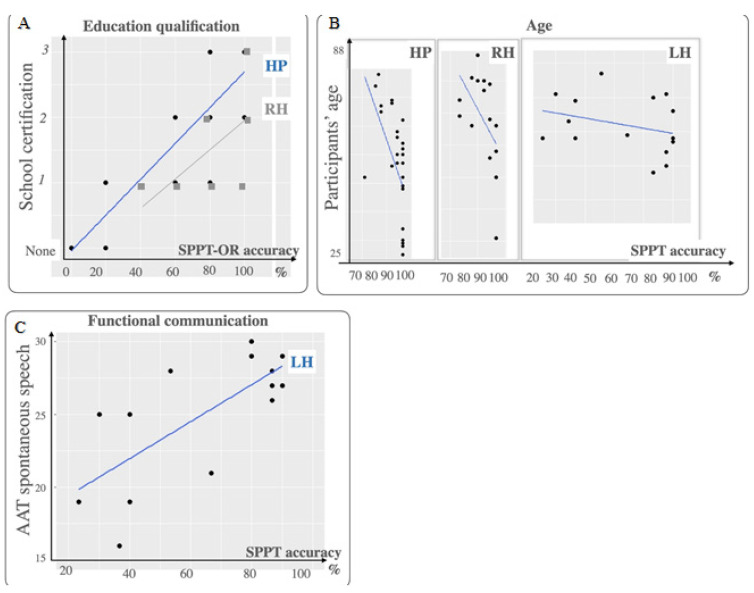
Predictive factors for NAVS-G SPPT accuracy. (**A**)**:** Results of the analysis group*sentence type*education relative to OR-SPTT (in y axis: leaving school certificates according to the German school system, see Table 1). HP and right hemispheric (RH) stroke patients with no leaving certificate performed worse than participants with Abitur (secondary high school) (see Table 6, *p* < 0.001 and = 0.003, respectively R^2^ = 0.947). (**B**): Results of the analysis group*sentence type*age: The older the HP and the RH patients (not the patient with aphasia), the worse the performance (*p* = 0.003 and = 0.043, respectively R^2^ = 0.655). (**C**): In patients with aphasia, it is functional communication, as reflected by the scores of the AAT spontaneous speech (*p* < 0.001, R^2^ = 0.18), which negatively relates to SPPT performance. The more difficulties in communication the patients had, the more difficulty they had in performing the SPPT.

**Table 1 brainsci-11-00474-t001:** Patients’ demographic and clinical data for each patient in the LHSP (left hemispheric stroke patients) group with aphasia and RHSP (right hemispheric stroke patients) group without aphasia, including educational qualification (Ed.q.)—according to the German school system: (1) no school leaving certificate, (2) junior high school certificate covering eight years (“Hauptschulabschluss”), (3) secondary high school leaving certificate covering ten years (“Realschulabschluss”), (4) Abitur; highest educational graduation level (Ed.g.): (1) no education degree, (2) completed apprenticeship degree, and (3) university degree; time post-onset (TPO; time between the stroke date and German language versions of the Northwestern Assessment of Verbs and Sentences (NAVS-G) and the Northwestern Anagram Test (NAT-G) testing); stroke severity scores: the National Institute for Health Stroke Scale (NIHSS), modified Rankin Scale, mRS), and necessity of care (Barthel Index, BI); Aachener Aphasia Test (AAT) spontaneous speech sample scores for: (S1), articulation and prosody (S2), automatized language (S3), semantic structure (S4), phonemic structure (S5), and syntactic structure (S6); Token Test (TT) scores (note that the TT was not administered to the RHSP group, because they generally show no deficits on this test [142]. The character * displays patients without Token Test data: patient n. 7 refused further testing and, for the patients number 13 and 14, the AAT’s testing was not available.

Patient	Gender	Age	Ed.q.	Ed.g.	TPO	Stroke’s Severity			Language (AAT Criteria)					Aphasia			
LHSP					(Months)	NIHSS	mRS	BI	S1	S2	S3	S4	S5	S6	TT	Severity	Type
1	m	71	2	2	9	5	3	80	4	5	5	3	5	3	44	Mild	Broca
2	w	69	3	2	6	1	1	100	4	5	5	4	3	4	48	Mild	Anomic
3	w	54	4	3	6	4	2	90	4	4	5	4	5	4	40	Mild	Anomic
4	w	71	3	2	5	0	1	100	4	5	5	4	5	5	50	Minimal	Residual
5	m	58	4	2	5	0	1	100	5	5	5	4	5	5	50	Minimal	Residual
6	m	48	2	2	6	0	1	100	5	5	5	5	4	5	50	Minimal	Residual
7	m	66	2	2	5	0	1	100	5	5	5	4	5	5	*	Minimal	Residual
8	m	50	4	3	50	2	2	100	5	4	5	4	4	5	45	Minimal	Residual
9	w	59	4	2	310	4	2	100	3	3	5	4	4	2	49	Minimal	Residual
10	m	58	3	2	26	2	1	100	2	3	5	3	4	2	27	Mild	Broca
11	w	63	3	2	109	7	3	60	1	3	4	4	3	1	47	Middle	Broca
12	m	58	3	2	34	2	1	100	5	5	5	3	5	5	46	Mild	Broca
13	w	77	2	2	0	1	1	100	3	3	5	3	5	2	*	Minimal	Residual
14	m	57	4	2	3	1	1	100	5	5	5	4	5	5	*	Minimal	Residual
15	m	70	4	3	6	0	1	100	5	5	5	4	5	5	45	Minimal	Residual
**Mean (SD)**		61.35 (8.18)	3.07 (0.81)	2.14 (0.40)	38.66 (77.54)	1.9 (2.08)	1.4 (0.72)	95.33 (10.87)	3.9 (1.21)	4.2 (0.87)	4.9 (0.25)	3.8 (0.54)	4.4 (0.71)	3.7 (1.41)	45 (6.43)		
***RHSP***																	
1	m	79	3	2	0	1	1	85	5	5	5	5	5	5			
2	m	69	2	2	0	2	2	80	5	5	5	5	5	5			
3	m	74	2	2	0	0	0	100	5	5	5	5	5	5			
4	w	80	2	2	0	2	1	60	5	5	5	5	5	5			
5	m	56	3	2	6	0	1	100	5	5	5	5	5	5			
6	m	77	2	2	19	1	1	100	5	5	5	5	5	5			
7	m	66	2	2	7	0	0	100	5	5	5	5	5	5			
8	w	58	4	3	5	0	0	100	5	5	5	5	5	5			
9	w	31	3	2	6	0	0	100	5	5	5	5	5	5			
10	m	50	4	3	6	0	1	95	5	5	5	5	5	5			
11	w	81	2	1	5	8	4	25	5	5	5	5	5	5			
12	w	68	2	2	0	0	0	100	5	5	5	5	5	5			
13	w	80	2	1	0	2	1	80	5	5	5	5	5	5			
14	w	66	2	2	0	0	1	90	5	5	5	5	5	5			
15	w	88	2	1	7	1	2	100	5	5	5	5	5	5			
**Mean (SD)**		68.2 (14.24)	2.47 (0.72)	1.93 (0.57)	4.07 (4.96)	1 (2.0)	1 (1.3)	87.67 (20.15)	5 (0.0)	5 (0.0)	5 (0.0)	5 (0.0)	5 (0.0)	5 (0.0)			

**Table 2 brainsci-11-00474-t002:** The NAVS-G and NAT-G structure.

Subtest	Number of Test Items	Description	Stimuli
VNT	22 + 2 examples	Verb naming (an action shown on a drawing)	5 obligatory one-argument verbs 5 obligatory two-argument verbs 5 optional two-argument verbs 2 obligatory three-argument verbs 5 optional three-argument verbs
VCT	22 + 2 examples	Verb comprehension (choosing the picture named by the examiner out of 4)	same as VNT
ASPT	32 + 2 examples	Active sentences production based on verbs with different argument structures. (All words needed for the sentences are given)	every verb appearing in VNT and VCT is tested in all its argument structures
NAT	30 + 2 examples	nonverbal production of sentences with different syntactic complexity by arranging printed word cards	5 active sentences 5 passive sentences 5 subject extracted Wh-question 5 object extracted Wh-question 5 subject relative sentences 5 object relative sentences
SPPT	30 + 3 examples	production of sentences with different syntactic complexity (primed by the sentence structure given by the examiner)	same as NAT
SVT	30 + 3 examples	comprehension of sentences with different syntactic complexity (choosing the correct picture out of two)	same as NAT

**Table 3 brainsci-11-00474-t003:** Descriptive statistics of the NAVS-G and NAT-G. Percentage mean accuracy (M) and standard deviation (SD) are provided by verb type for subtests assessing verb argument structure (VAS) processing for each participant group (RHSP: right hemispheric stroke patients without aphasia; LHSP: left hemispheric stroke patients with aphasia) and by sentence type and canonical/non-canonical word order for subtests assessing sentence processing. Abbreviations: Ob = obligatory; Op = optional; SW = subject Wh-question; OW object Wh-questions; SR = subject relative; OR = object relative. VNT = Verb Naming Test, VCT = Verb Comprehension Test, ASPT = Argument Structure Production Test, NAT = Northwestern Anagram Test, SPPT = Sentence Production Priming Test, SCT = Sentence Comprehension Test.

HP: Healthy Participants without Aphasia							RHSP Participants without Aphasia							LHSP Participants with Aphasia						
	**VNT**		**VCT**		**ASPT**			**VNT**		**VCT**		**ASPT**			**VNT**		**VCT**		**ASPT**	
	M	SD	M	SD	M	SD		M	SD	M	SD	M	SD		M	SD		SD		SD
ob1	100	0.00	100	0.00	100	0.00	ob1	100	0.00	100	0.00	100	0.00	ob1	98.60	5.16	100	0.00	99.33	2.58
op2	99.26	3.85	99.23	3.85	100	0.00	op2	100	0.00	100	0.00	100	0.00	op2	82.70	7.04	100	0.00	97.48	6.97
ob2	99.26	3.85	100	0.00	100	0.00	ob2	98.67	5.16	100	0.00	100	0.00	ob2	97.30	14.86	100	0.00	97.07	8.07
op3	100	0.00	100	0.00	100	0.00	op3	100	0.00	100	0.00	100	0.00	op3	89.30	14.86	98.67	5.16	95.76	10.68
ob3	100	0.00	100	0.00	100	0.00	ob3	100	0.00	100	0.00	100	0.00	ob3	63.30	35.19	100	0.00	90.00	19.21
mean	99.67	1.21	99.83	0.87	100	0.00	mean	99.67	1.17	100	0.00	100	0.00	mean	89.40	8.18	99.70	1.17	96.48	8.32
	**NAT**		**SPPT**		**SCT**			**NAT**		**SPPT**		**SCT**			**NAT**		**SPPT**		**SCT**	
	M	SD	M	SD	M	SD		M	SD	M	SD	M	SD		M	SD	M	SD	M	SD
Active	100	0.00	100	0.00	100	0.00	Active	100	0.00	100	0.00	100	0.00	Active	96.92	7.51	97.33	10.33	98.67	5.16
Passive	100	0.00	99.26	3.84	99.26	3.84	Passive	100	0.00	100	0.00	100	0.00	Passive	75.38	39.29	65.33	41.03	88.00	22.42
SW	100	0.00	99.26	3.84	99.26	3.84	SW	100	0.00	97.33	7.04	98.67	5.16	SW	90.77	22.53	82.66	26.04	96.00	11.21
OW	100	0.00	97.78	6.41	99.26	3.84	OW	100	0.00	88.00	14.74	98.67	5.16	OW	76.92	33.51	66.67	41.86	89.33	22.51
SR	100	0.00	97.04	9.12	100	0.00	SR	98.67	5.16	94.67	14.07	100	0.00	SR	69.23	45.18	54.67	38.89	90.67	16.68
OR	100	0.00	80.74	32.57	98.52	7.70	OR	90.67	19.81	77.33	21.02	96.00	11.21	OR	60.00	42.43	25.33	24.46	82.67	19.81
C	100	0.00	98.77	3.22	99.75	1.28	C	99.56	1.72	97.33	4.91	99.56	1.72	C	85.64	22.58	78.22	22.03	95.11	9.25
non-C	100	0.00	92.59	12.59	99.01	5.13	non-C	96.89	6.60	88.44	11.12	98.22	4.69	non-C	70.77	36.47	52.44	32.16	86.67	14.70
mean	100	0.00	95.68	7.21	99.38	2.62	mean	98.22	3.96	92.89	7.11	98.89	3.00	mean	78.21	29.02	65.33	25.19	90.89	8.77

**Table 4 brainsci-11-00474-t004:** Fixed effects results of the logistic regression analysis conducted on VNT-G accuracy in the LHSP group with aphasia (A) and between participants groups (B). Individual model results for the seven predictors: (1) verb argument optionality (VAO), (2) verb argument number (VAN), (3) verb type (VT), (4) AAT global scores (AAT.gs), (5) AAT syntax (AAT.sy), (6) Verb frequency, and (7) age. The VAN and VT models were refitted including age as a covariate. Multiple comparisons results are reported when significant. (B) Significant results of the between population groups fixed effects analysis (for complete results please see Appendix A). Abbreviation: LH = LHSP, RH = RHSP. Note: z-values are provided for continuous variables or categorical variables with only two levels; for categorical variables with 3 or more levels, χ^2^ values are reported. The character ^ indicates the effects that are statistically significant and have sufficient power (>80%) and R-squared >0.26. Beta estimates and standard errors are not provided for categorical variables with three or more levels, because the overall significance of the predictor was calculated by comparing models with and without it. Beta estimates and standard errors are provided for pairwise comparisons.

A. NAVS-G VNT: LHSP with Aphasia							
***Fixed Effects***	**R^2^**	***Estimate***	***SE***	***p***	***z***	***χ*^2^**	***Power***
*Single models*							
**VAN**	0.193			0.047		6.106	70%
**VAO**	0.054	1.087	0.738	0.140	1.474		29%
**VT ^**	0.297			0.002		17.46	97%
**AAT.gs**	0.592	0.004	0.002	0.036	2.101		0%
**AAT.sy**	0.048	0.374	0.147	0.011	2.555		64%
**Verb freq ^**	0.157	−1.043	0.376	0.006	−2.774		82%
**Age**	0.062	−0.072	0.029	0.013	−2.478		63%
*Adjusted models*							
**VAN + Age**	0.248						
VAN				0.047	6.131		74%
Age		−0.072	0.029	0.013	−2.48		
**VT + Age ^**	0.357						
VT				0.001		17.582	98%
ob2 < ob1		−2.845	1.121	0.028	2.54		
ob3 < ob1		−4.001	1.197	0.004	3.34		
op2 > ob2		2.127	0.869	0.029	2.447		
op2 > ob3		3.283	0.964	0.004	3.406		
op3 > ob3		2.098	0.767	0.021	2.735		
Age		−0.072	0.029	0.013	−2.493		
**B. Between groups**							
***Fixed effects***	**R^2^**	***Estimate***	***SE***	***p***	***z***	***χ*^2^**	***Power***
Group	0.238			0.000		29.33	
HP > RH		0.416	0.623	0.504	0.668		
HP > LH ^		2.478	0.470	0.000	5.275		
RH > LH ^		2.062	0.510	0.000	4.046		

**Table 5 brainsci-11-00474-t005:** Fixed effects results of the logistic regression analysis conducted on argument structure production test (ASPT)-G accuracy in the LHSP with aphasia. Individual model results for the seven predictors: 1 verb argument optionality (VAO), 2 verb argument number (VAN), 3 verb type (VT), 4 NIHSS, 5 mRS and 6 BI. The VAN and VT models were refitted including NIHSS as a covariate. Multiple comparisons results are reported when significant (for complete results please see Appendix A). Note: z-values are provided for continuous variables or categorical variables with only two levels; for categorical variables with 3 or more levels, χ^2^ values are reported. The character ^ indicates the effects that are statistically significant and have sufficient power (>80%) and R-squared >0.26. Beta estimates and standard errors are not provided for categorical variables with three or more levels, because the overall significance of the predictor was calculated by comparing models with and without it. Beta estimates and standard errors are however provided for pairwise comparisons.

NAVS-G ASPT: LHSP with Aphasia							
***Fixed Effects***	***R^2^***	***Estimate***	***SE***	***p***	**z**	***χ*^2^**	***Power***
***Single predictor models***							
**VAN**	0.039			0.000	15.919		66%
**VAO**	0.006	−0.882	0.637	0.166	−1.386		24%
**VT**	0.039			0.002		16.896	78%
**NIHSS**	0.241	−0.924	0.418	0.027	−2.209		61%
**mRS**	0.280	−2.906	1.212	0.016	−2.938		69%
**BI ^**	0.281	0.170	0.063	0.006	2.721		84%
***Adjusted models***							
**VAN + NIHSS**	0.327						
VAN ^				0.000		16.026	87%
2Arg < 1Arg		−3.586	1.220	0.010	2.939		
3Arg < 1Arg		−2.261	1.174	0.054	1.925		
3Arg > 2Arg		1.325	0.525	0.017	2.523		
NIHSS		−0.952	0.431	0.027	−2.211		
**VT + NIHSS**	0.329						
VT ^				0.002		16.896	87%
ob3 < ob1		−4.205	1.402	0.024	2.998		
op3 < ob1		−3.416	1.210	0.024	2.823		
NIHSS		−0.9556	0.4338	0.027	−2.203		

**Table 6 brainsci-11-00474-t006:** Fixed effects results of the logistic regression analysis conducted on NAVS-G SPPT accuracy in all participants. A. Results of the within group analysis. Individual model results for the eight predictors: 1 canonicity, 2 canonicity without relatives (i.e., A + SW > P + OW), 3 sentence type (ST), and other covariates. Results of the single models 1–3 refitted by the best predictors: AAT spontaneous speech (AAT.ss) and NIHSS in the LHSP group and education qualification (Ed.q.) in the HP and RHSP groups. Post hoc results of the adjusted models are reported, if significant. B. Significant results of the between population groups fixed effects analysis. Note: z-values are provided for continuous variables or categorical variables with only 2 levels; for categorical variables with 3 or more levels, *χ*^2^ values are reported. The character ^ indicates the effects that are statistically significant and have sufficient power (>80%) and R^2^ > 0.26. Abbreviation: Fixed eff. = fixed effects, Est. = Estimates, SE = Standard error, Pw = Power, LH = LHSP, RH = RHSP, C = canonical, nC = non canonical, ST = sentence type, A = active, SW = subject Wh-question, OW object Wh-questions, SR = subject relative, OR = object relative, Ed.q. = educational qualification, Ed.g. = highest educational graduation level.

NAVS-G SPPT in LHPP with Aphasia							NAVS-G SPPT in RHSP without Aphasia							NAVS-G SPPT in HP						
***Fixed eff.***	***Est.***	***SE***	***p***	***χ*^2^**	**z**	***Pw***	***Fixed eff.***	***Est.***	***SE***	***p***	***χ*^2^**	**z**	***Pw***	***Fixed eff.***	***Est.***	***SE***	***p***	***χ*^2^**	**z**	***Pw***
*Single predictor models*							*Single predictor models*							*Single predictor models*						
**Canonicity (R^2^ = 0.129)**							**Canonicity (R^2^ = 0.073)**							**Canonicity (R^2^ = 0.119)**						
Canonicity	2.318	0.661	0.000		3.504	98%	Canonicity	1.438	0.780	0.065		1.844	49%	Canonicity	1.85	0.93	0.0465		1.99	56%
**C. without relatives (R^2^ = 0.108)**							**C. without relatives (R^2^ = 0.04)**							**C. without relatives (R^2^ = < 0.001)**						
A + SW > P + OW	2.618	0.551	<0.001		4.748	21%	A + SW > P + OW	1.017	1.086	0.349		0.936	19%	A + SW > P + OW	0.003	0.826	0.997		0.003	6%
**Sentence type (R^2^ = 0.394)**							**Sentence type (R^2^ = 0.234)**							**Sentence type (R2 = 0.239)**						
ST			<0.001	82.395		100%	ST			0.018	13.672		82%	ST			0.000	42.442		100%
**NIHSS (R^2^ = 0.09)**							**Ed. (R^2^ = 0.162)**							**Ed.** **q. (R2= 0.188)**						
NIHSS	−0.466	0.205	0.023		−2.277	66%	Ed.q.	1.568	0.567	0.006		2.768	80%	Ed.q.	1.441	0.266	0.000		5.427	100%
**AAT.ss (R^2^ = 0.177)**							**Ed. Grad. (R^2^ = 0.039)**							**Ed. Grad. (R^2^ = 0.114)**						
AAT.ss	0.320	0.083	0.000		3.867	96%	Ed.grad	0.939	0.613	0.126		1.531	39%	Ed.grad	1.488	0.468	0.001		3.181	89%
**AAT.sy (R^2^ = 0.135)**							**Age (R^2^ = 0.103)**							**Age (R^2^ = 0.243)**						
AAT.sy	0.845	0.278	0.002		3.038	85%	Age	−0.063	0.029	0.029		−2.181	70%	Age	−0.095	0.029	0.001		−3.314	96%
**Aph.se (R^2^ = 0.168)**																				
Aph.se	1.125	0.337	0.001		3.338	0%														
**Token Test (R^2^ = 0.111)**																				
TT	0.178	0.080	0.025		2.236	0%														
*adjusted models*							*adjusted models*							*adjusted models*						
**Canonicity + AAT.ss (R^2^ = 0.307)**							**Canonicity + Ed.q. (R^2^ = 0.234)**							**Canonicity + Ed.q. (R^2^ = 0.28)**						
Canonicity	2.318	0.661	0.000		3.506	96%	Canonicity	1.432	0.775	0.065		1.847	43%	Canonicity	1.790	0.902	0.047		1.985	53%
AAT.ss	0.320	0.083	0.000		3.872		Ed.q.	1.562	0.564	0.00		2.767		Ed.q.	1.437	0.265	0.000		5.429	
**Canonicity without rel. + AAT.ss (R^2^ = 0.449)**							**C. without rel. + Ed.q. (R^2^ = 0.140)**							**C. without rel. + Ed.q. (R^2^ = 0.059)**						
A + SW > P + OW	2.6481	0.5583	0.000		4.743	99%	A + SW > P + OW	1.027	1.091	0.346		0.942	16%	A + SW > P + OW	−0.005	0.826	0.995		−0.006	7%
AAT.ss	0.5021	0.164	0.002		3.062		Ed.q.	1.150	0.718	0.109		1.602		Ed.q.	0.533	0.447	0.233		1.193	
**Sentence type + AAT.ss (R^2^ = 0.569)**							**Sentence type + Ed.q. (R^2^ = 0.376)**							**Sentence type + Ed.q. (R^2^ = 0.507)**						
ST			0.000	82.377		100%	ST			0.018	13.671		92%	ST			0.000	42.457		100%
OW < SW	−1.407	0.502	0.006		−2.805		OR < SR	−1.930	1.000	0.179		−1.930		OR < SR	−2.665	0.677	0.000		−3.935	
OR < SR	−2.044	0.466	0.000		−4.390		OR < P	−3.604	1.359	0.040		−2.652		OR < P	−4.113	1.106	0.000		−3.720	
SR < A	−4.807	0.861	0.000		−5.583		OR < OW	−1.241	0.957	0.325		−1.296		OR < OW	−3.381	0.845	0.000		−4.001	
SW < A	−2.480	0.845	0.004		−2.935		nOR < OR	−9.599	3.176	0.025		−3.023		nOR < OR	−13.484	2.215	0.000		−6.087	
OR < P	−2.858	0.501	0.000		−5.701		nRel < Rel	−1.919	0.826	0.020		−3.250		nRel < Rel	−2.898	0.730	0.000		−3.969	
SR < SW	−2.327	0.517	0.000		−4.500		Ed.q.	1.566	0.566	0.006		2.769		Ed.q.	1.426	0.263	0.000		5.417	
nOR < OR	−12.239	1.685	0.000		−7.264															
AAT.ss	0.329	0.085	0.000		3.866															
**Canonicity + NIHSS (R^2^ = 0.219)**																				
Camonicity	2.316	0.661	0.000		3.505	96%														
NIHSS	−0.465	0.204	0.023		−2.281															
**Canonicity without relatives + NIHSS (R^2^ = 0.274)**																				
A + SW > P + OW	2.629	0.554	0.000		4.742	100%														
NIHSS	−0.783	0.420	0.062		−1.864															
**Sentence type + NIHSS (R^2^ = 0.482)**																				
ST			< 0.001	69.185		100%														
P < A	−3.974	0.845	0.000		−4.702															
OW < SW	−1.401	0.500	0.006		−2.800															
OR < SR	−2.056	0.467	0.000		−4.403															
SR < A	−4.785	0.858	0.000		−5.574															
NIHSS	−0.477	0.210	0.023		−2.276															
**NAVS-G SPPT: Between population-grpups results**																				
***Fixed eff.***	***Est.***	***SE***	***p***	***χ*^2^**	**z**	***Pw***	***Fixed eff.***	***Est.***	***Est.***	***p***	***χ*^2^**	**z**	***Pw***	***Fixed eff.***	***Est.***	***SE***	***p***	***χ*^2^**	**z**	***Pw***
*Single predictor models*																				
**Group (R^2^ = 0.229)**							**Group Sentence type (R^2^ = 0.441)**							**group sent. type age (R^2^ = 0.654)**						
Group			0.000	35.736		100%	Group			0.000	4.021			Group			0.000	38.928		
HP > RH	1.078	0.625	0.084		1.727		ST			0.013	22385			ST			0.000	83.104		
HP > LH ^	3.857	0.620	0.000		6.226		*SWQ*							Age	−0.066	0.024	0.006		−2.728	
RH > LH ^	2.779	0.650	0.000		4.278		Group			0.051	5.934		43%	Group age ST			0.0218	23.773		
*models with interactions*							HP > LH	2.955	1.128	0.026		2.621		ST age			0.032	12.238		
**Group canonicity (R^2^ = 0.318)**							*OWQ*							Group age			0.023	0.397		
Group			0.000	4.021			Group			0.013	8.691		66%	HP	−0.095	0.028	0.003		−3.314	98%
Canonicity			0.000	12.526			HP > RH	3.010	1.406	0.048		2.141		RH	−0.062	0.002	0.043		−2.181	62.5%
Group Canonicity			0.512	0.774			HP > LH	4.905	1.479	0.003		3.316		**Group Sentence type Education (R^2^= 947)**						
**C. without relatives (R^2^ = 0.143)**							*SR*							Group			0.000	47.131		
Group			0.000	15.269			Group			0.000	24.200		94%	Sentence type			0.000	75.390		
Canonicity without relatives			0.000	17.310			HP > LH	5.489	1.486	0.001		3.693		Ed.q.	1.423	0.388	0.000		3.671	
*A + SWQ*						56%	RH > LH	4.980	1.631	0.003		3.054		ST Ed.q.			0.067	1.300		
HP > LH	2.450	0.901	0.020		2.718		*OR*							Group ST Ed.q.			0.000	36.714		
*P + OWQ*						97%	Group			0.000	3.263		100%	HP			0.000	4.667		100%
HP > RH	2.426	1.104	0.028		2.197		HP > LH	3.771	0.794	0.000		4.749		RH			0.003	2.323		55%
HP > LH	4.705	1.086	0.000		4.333		RH > LH	3.205	0.799	0.000		4.011								
RH > LH	2.280	0.986	0.028		2.312															
Group Canonicity without relatives			0.029	7.071																

**Table 7 brainsci-11-00474-t007:** Fixed effects results of the logistic regression analysis conducted on NAVS-G SCT accuracy. A. Results in the LHSP group with aphasia. Individual model results for the eight predictors: 1 canonicity, 2 canonicity without relatives (i.e., A + SW > P + OW), 3 sentence type, and other covariates. Results of the single models 1–3 refitted by the best predictors: AAT spontaneous speech (AAT.ss). Post hoc results of the adjusted models are reported, if significant. B. Significant results of the between population groups fixed effects analysis ^ Note: z-values are provided for continuous variables or categorical variables with only 2 levels; for categorical variables with 3 or more levels, *χ*^2^ values are reported. The character ^ indicates the effects that are statistically significant and have sufficient power (>80%) and R^2^ > 0.26.

NAVS-G SCT Results in LHSP with Aphasia							
***Fixed Effects***	***R^2^***	***Estimate***	***SE***	***p***	***χ*^2^**	**z**	***Power***
**single predictor models**							
**Canonicity ^**	0.088	1.329	0.466	0.004		2.851	83%
**Canonicity without relatives**	0.092	1.424	0.599	0.018		2.376	69%
**Sentence type ^**	0.19			0.004	17.211		100%
**NIHSS**	0.058	−0.254	0.121	0.036		−2.098	67%
**BI**	0.047	0.044	0.022	0.042		2.032	58%
**AAT.ss ^**	0.096	0.161	0.053	0.003		3.005	91%
**AAT.sy**	0.055	0.368	0.187	0.050		1.961	53%
**Aph.se**	0.108	0.701	0.270	0.009		2.599	0%
**adjusted models**							
**Canonicity + AAT.ss**	0.185						
Canonicity ^		1.332	0.467	0.004		2.853	85%
AAT.ss		0.161	0.053	0.003		3.015	
**Canonicity without relatives + AAT.ss**	0.286						
Canonicity		1.435	0.602	0.017		2.385	72%
AAT.ss		0.232	0.082	0.005		2.821	
**Sentence type + AAT.ss**	0.287						
Sentence type ^				0.004	17.574		97%
OR < non-OR		−5.812	1.700	0.006		−3.418	
Rel < non-Rel		−1.145	0.454	0.012		−2.519	
AAT.ss		0.162	0.053	0.002		3.026	
**NAVS-G SCT: between group results**							
***Fixed effects***	***R*^2^**	***Estimate***	***SE***	***p***	***χ*^2^**	**z**	***Power***
**Single predictor model**							
Group	0.213			0.000	17.401		98.00%
HP > LHSP		−3.078	0.770	0.000		3.996	
RHSP > LHSP		−2.357	0.817	0.006		2.885	

**Table 8 brainsci-11-00474-t008:** Fixed effects results of the logistic regression analysis conducted on NAT-G accuracy in LHSP group with aphasia (A). Individual model results for the eight predictors: 1 canonicity (C.), 2 canonicity without relatives (i.e., A + SW > P + OW), 3 Sentence type. refitted by the best predictors: AAT spontaneous speech (AAT.ss). Post hoc results of the adjusted models are reported, if significant. B. Significant results of the between population groups fixed effects analysis ^ Note: z-values are provided for continuous variables or categorical variables with only 2 levels; for categorical variables with 3 or more levels, *χ*^2^ values are reported. * indicates the effects that are statistically significant and had sufficient power (>80%) and R^2^ > 0.26.

A.NAT-G Results in LHSP with Aphasia							
***Fixed effects***	***R*^2^**	***Estimate***	***SE***	***p***	***χ*^2^**	**z**	***Power***
**Single predictor models**							
**Canonicity**	0.066	2.228	0.652	0.001		3.416	96%
**Canonicity without relatives**	0.000	2.883	0.652	0.000		4.425	91%
**Sentence type**	0.194			0.000	39.635		99%
**NIHSS**	0.235	−1.042	0.468	0.026		−2.227	77%
**AAT.sy * ^**	0.306	1.630	0.551	0.003		2.958	89%
**Aph.se**	0.287	1.805	0.654	0.006		2.761	0%
**Token Test**	0.259	0.319	0.126	0.011		2.530	0%
**AAT.ss * ^**	0.41	0.613	0.157	0.000		3.916	99%
**adjusted models**							
**Canonicity + AAT.ss** 0.485							
Canonicity		2.273	0.663	0.001		3.426	99%
AAT.ss		0.630	0.161	0.000		3.921	
**Canonicity without relatives + AAT.ss** (*R*^2^ 0.567)							
Canonicity without relatives		2.861	0.637	0.000		4.489	100%
AAT.ss		0.627	0.182	0.001		3.444	
**Sentence type + AAT.ss**	0.615						
Sentence type				0.000	4.124		100%
P < A		−4.137	1.035	0.000		3.996	
OW < SW		−2.258	0.780	0.005		2.895	
SR < A		−5.067	1.073	0.000		4.722	
OR < P		−2.169	0.730	0.005		2.972	
OR < OW		−2.403	0.743	0.002		3.236	
SR < SW		−3.421	0.832	0.000		4.113	
OR < non-OR		−1.472	2.435	0.000		4.301	
Rel < Non-Rel		−3.104	0.712	0.000		4.261	
AAT.ss		0.679	0.171	0.000		3.963	
**between group analyses**							
***Fixed effects***	***R^2^***	***Estimate***	***SE***	***p***	***χ*^2^**	**z**	***Power***
**Single predictor models**							
Group	0.196						
Group				0.001	13.309		88.00%
HS > RHSP		−1.431	1.024	0.162		1.398	
HS > LHSP		−3.787	0.981	0.000		3.861	
RHSP > LHSP		−2.355	1.041	0.036		2.262	
**models with interactions**							
**Group * Canonicity**	0.199						
Group				0.002	12.902		
Canonicity				0.001	1.756		
Group * Canonicity				0.150	3.792		
**Group * Canonicity without relatives**	0.120						
Group				0.067	5.396		
Canonicity without relatives				0.001	11.093		
Group * Canonicity without relatives				0.018	80.031		
*Non-Canonical*							
Group	0.249			0.002	12.692		77.00%
HS > LHSP		4.834	1.281	0.000	3.774		
RHSP > LHSP		2.908	1.293	0.037	2.250		
**Group * Sentence type**	0.204						
Group				0.010	9.237		
Sentence type				0.000	3.745		
Group * Sentence type				0.059	17.763		

**Table 9 brainsci-11-00474-t009:** Errors produced by LHSP group participants when performing NAVS-G and NAT-G.

Test	Error Type in Patients with Aphasia	Number	%
VNT	Semantic paraphasia with a verb with:	24	71
	A. the same argument (e.g., schiebento push—instead of ziehen—to pull)	17	
	B. a lower argument when the target verb was 3 Ob and 3 Op (e.g., holen—to get instead of geben—to give)	7	
	Verb omission	6	18
	Phonematic paraphasia	2	6
	Substitution by nouns	1	3
	Perseveration	1	3
ASPT	Missing verb’s conjugation together with wrong verb’s position (e.g., Der Mann Brief schicken—The man letter send)	26	56
	The omission of one given argument:	6	13
	A. of the agents (e.g., Retten die Frau—Saves the woman)	1	
	B. of the goal (e.g., Mann stellt Schachtel—Man puts box)	1	
	C. of the patients (e.g., Der Mann schreibt einen Brief, instead of Der Mann schreibt der Frau einen Brief—The man writes a letter, the word woman is missing)	4	
	Wrong order of arguments in sentences with 3 arguments (e.g., Der Mann schickt einen Brief der Frau—The man sends a letter the woman. (In German the dative object is always before the accusative object when this one is preceded by an undefined article:The correct sentence in German is: Der Mann schickt der Frau einen Brief)).	5	11
	Missing/wrong conjugation with right verb position (e.g., Hund beißen Katze—Dog bite cat)	3	6
	Use of a wrong, not given verb (e.g., Das Baby strampelt statt krabbelt—The baby is kicking instead of crawling)	3	6
	Perseveration	1	2
	Omission of the whole sentence	1	2
	Role reversal (e.g., Die Frau rettet den Mann—The woman is saving the man, while the picture showed the opposite)	2	4
SPPT	Incorrect word order	96	64
	Passive subordinate clauses instead of an OR sentences	23	15
	Canonical instead of non-canonical	21	14
	Reveal of roles	4	3
	Verb omission (SR/OR)	3	2
	Use of wrong personal pronouns	3	2

**Table 10 brainsci-11-00474-t010:** Correlations across all subtests.

Results of the Pearson-Correlation Analysis across All Subtests of the NAVS-G and NAT-G									
	**VNT**			**VCT**			**ASPT**		
	*p*-value	r	CI	*p*-value	r	CI	*p*-value	r	CI
VNT		1		0.029	0.564	0.07|0.84	0.007	0.661	0.23|0.88
VCT	0.029	0.564	0.07|0.84		1		0.000	0.878	0.67|0.96
ASPT	0.007	0.661	0.23|0.88	0.000	0.878	0.67|0.96		1	
NAT	0.096	0.481	−0.1|0.82	0.276	0.327	−0.27|0.74	0.115	0.459	−0.12|0.81
SPPT	0.104	0.436	−0.1|0.78	0.253	0.315	−0.24|0.71	0.069	0.482	−0.04|0.8
SCT	0.041	0.533	0.03|0.82	0.094	0.449	−0.08|0.78	0.017	0.605	0.14|0.85
NAVS-G and NAT-G	0.008	0.656	0.22|0.87	0.010	0.640	0.19|0.87	0.001	0.777	0.44|0.92
	**NAT**			**SPPT**			**SCT**		
	*p*-value	r	CI	*p*-value	r	CI	*p*-value	r	CI
VNT	0.096	0.481	−0.1|0.82	0.104	0.436	−0.1|0.78	0.041	0.533	0.03|0.82
VCT	0.276	0.327	−0.27|0.74	0.253	0.315	−0.24|0.71	0.094	0.449	−0.08|0.78
ASPT	0.115	0.459	−0.12|0.81	0.069	0.482	−0.04| 0.8	0.017	0.605	0.14|0.85
NAT		1		0.000	0.881	0.64|0.96	0.000	0.845	0.55|0.95
SPPT	0.000	0.881	0.64|0.96		1		0.000	0.904	0.73|0.97
SCT	0.000	0.845	0.55|0.95	0.000	0.904	0.73|0.97		1	
NAVS-G and NAT-G	0.000	0.848	0.56|0.95	0.000	0.816	0.52|0.94	0.000	0.896	0.71|0.96

**Table 11 brainsci-11-00474-t011:** Comparison between the NAVS-G and NAT-G subsets for participants with aphasia.

B. Results of the Comparison between the Subtests Using Wilcoxon Tests.																		
	VNT			VCT			ASPT			NAT			SPPT			SCT		
	*p*	Z	*r*	*p*	Z	*r*	*p*	Z	*r*	*p*	Z	*r*	*p*	Z	*r*	*p*	Z	*r*
VNT				0.002	−3.078	0.795	0.003	−2.976	0.768	0.346	0.942	0.261	0.003	2.954	0.763	0.002	−3.078	0.854
VCT	0.002	3.078	0.795				0.043	2.023	0.522	0.007	−2.675	0.742	0.001	3.415	0.882	0.003	−2.956	0.762
ASPT	0.003	2.976	0.768	0.043	−2.023	0.522				0.009	−2.601	0.721	0.001	3.411	0.881	0.015	−2.443	0.631
NAT	0.346	0.942	0.261	0.007	−2.675	0.742	0.009	−2.601	0.721				0.005	2.836	0.787	0.065	−1.843	0.511
SPPT	0.003	−2.954	0.763	0.001	−3.415	0.882	0.001	−3.411	0.881	0.005	−2.836	0.787				0.001	−3.413	0.881
SCT	0.002	3.078	0.854	0.003	−2.956	0.763	0.015	−2.442	0.631	0.065	1.843	0.511	0.001	3.413	0.881			

## Data Availability

All the data from random end fixed effect analysis presented in this study are reported in Suplementary materials and Appendix A. Individual raw data (including fMRI) are available on request due to ethical restrictions

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
