# Peer review of "German Language Adaptation of the NAVS (NAVS-G) and of the NAT (NAT-G): Testing Grammar in Aphasia"

_brainsci, 2021, doi:10.3390/brainsci11040474_

Round 1
Reviewer 1 Report
The article tests a German version of a grammatical diagnosis tools for aphasia. Based on small samples from people with aphasia, non-aphasic people with right-hemisphere damage, and controls, the authors (rightfully, in my view) conclude that the tests have the potential to coontribute to diagnosis in German speaking communities by improving profiling of grammatical capacity.
This is potentially impactful work, and much care has been put into the design of test materials and the profiling of participants. However, I see subtantial flaws in the manuscript which need to be addressed.
1. The introduction is, in my view, not appropriate for the work. It starts with a explanation of Chomskyan generative grammar with sections that do not benefit the paper (e.g. the entire first paragraph). The paper by Chomsky, Hauser and Fitch is cited even though it has no relevance for aphasia (Chomsky would agree - he himself considers evidence from aphasia not relevant to his work). While linguistic theory is important for understanding the design of the test materials, UG and principles and parameters are distractions from the actual contribution of the paper. Important topics, such as the importance of grammatical tests for research and clinical work, on the other hand, receive too little attention (the paragraph starting at page 3, line 122, should be at the heart of the intro).
The intro also contains many claims which are either unsupported by references or which need to be elaborated. i will only pick examples and adivce that the authors go through the manuscript again and apply the academic rigour the reader will expect.
Page 2, line 57: "Several fMRI studies [...]" I would elaborate or remove.
Page 3, line 127: "The most used test [...]" No reference provided.
Next sentence : "provides an excellent clinical profile" Not substantiated.
Page 3, line 144: "Currently, it has been further demonstrated" Reference missing
Page 6, line 263: "Several types of evidence..." References missing.
2. Many format errors. Acronyms introduced without explanation, then used inconsistently later. Number format switching from German to English. Missing spaces and capiitalization.
3. Participants: The study selected participants carefully and under strict criteria. I found it strange, however, to see someone as young as 25 in the healthy control group. The MWU test shows no significant difference in age, but that may be misleading since MWU looks and ranks and extreme values may get underestimated. Also, please explain inclusion and exclusion criteria since they exclude many with aphasia.
4. Page 7. Much of the MoCA is verbal (incl. the abstraction task). Why was it used?
5. Please report all effect sizes. I would also appreciate power calculations for the most important effects and p-values close to the sig. threshold. In my view the latter are too readily dismissed given the small samples.
6. I cannot evaluate the discussion without the requested additions to the data reporting. However, the discussion about education and grammatical knnowledge should include Street & Dabrowska's work on the topic.
Author Response
The article tests a German version of a grammatical diagnosis tools for aphasia. Based on small samples from people withaphasia, non-aphasic people with right-hemisphere damage, and controls, the authors (rightfully, in my view) conclude that the tests have the potential to contribute to diagnosis in German speaking communities by improving profiling of grammatical capacity.
This is potentially impactful work, and much care has been put into the design of test materials and the profiling of participants.
However, I see subtantial flaws in the manuscript which need to be addressed.
The introduction is, in my view, not appropriate for the work. It starts with a explanation of Chomskyan generative grammar with sections that do not benefit the paper (e.g. the entire first paragraph). The paper by Chomsky, Hauser and Fitch is cited even though it has no relevance for aphasia (Chomsky would agree - he himself considers evidence from aphasia not relevant to his work). While linguistic theory is important for understanding the design of the test materials, UG and principles and parameters are distractions from the actual contribution of the paper. Important topics, such as the importance of grammatical tests for research and clinical work, on the other hand, receive too little attention (the paragraph starting at page 3, line 122, should be at the heart of the intro).
We agree with the review that specific Chomsky’s aspects of the generative theory of grammar are not relevant, and have subsequently reviewed the introduction accordingly, with emphasis on the importance of grammatical tests for research and clinical work (in particular at page paragraph starting at page 5, line 214).
The intro also contains many claims which are either unsupported by references or which need to be elaborated. I will only pick examples and adivce that the authors go through the manuscript again and apply the academic rigour the reader will expect.
Page 2, line 57: "Several fMRI studies [...]" I would elaborate or remove.
We removed this sentence.
Page 3, line 127: "The most used test [...]" No reference provided.
Page 3, line 144: "Currently, it has been further demonstrated" Reference missing
We are desolated for missing the references. They are now all included on
Page 3, line 157
The standard diagnostic battery for chronic aphasia in Europe is the Aachener Aphasia Test (AAT) (Bastiaanse, Hurkmans, & Links, 2006; Gialanella, 2011; Greener, Enderby, & Whurr, 2001; Henseler, Regenbrecht, & Obrig, 2014; Huber, Willmes, Poeck, Van Vleymen, & Deberdt, 1997; Luzzatti et al., 1994; Willmes, 1985; Willmes, Poeck, Weniger, & Huber, 1983).
Page 4, line 181
In addition, it has been further demonstrated that the obligatory (obl) or optional (opt) status of verbal arguments - i.e., whether a verb requires the arguments involved in its lexical representation or allows some arguments to be omitted (i.e., they are optional) - may contribute to verb and sentence deficits in aphasia (Cho-Reyes & Thompson, 2012; Shapiro & Levine, 1990; Shapiro et al., 1987; Thompson et al., 2012).
Page 6, line 263: "Several types of evidence..." References missing.
This sentence was removed.
Many format errors. Acronyms introduced without explanation, then used inconsistently later. Number format switching from German to English. Missing spaces and capiitalization.
We are sorry for these mistakes. The actual manuscript has been carefully revised.
Participants: The study selected participants carefully and under strict criteria. I found it strange, however, to see someone as young as 25 in the healthy control group. The MWU test shows no significant difference in Also, please explain inclusion and exclusion criteria since they exclude many with aphasia.
The study selected participants between 18-88 years (we regret a mistake in the original manuscript, as we wrote 18-80). The rationale was that the study aimed to test the effect of age on syntactic ability in adults. Therefore, we had to include adult participants of every age, including 25-year-old subjects. Moreover, although our patients were older, it is also possible that also younger adults may suffer from a stroke. Sultan et al. 2013 [1] cited an incidence rate per 100,000 persons, 2.6/2.7 (men and women) among 18–24-year-olds and 12 (men and women) among 25–34 year old.
We note, however, that our new analyses, which were conducted using mixed-effects logistic regression, exclude the possibility that any of our main results are driven by the performance of single individuals. This is an advantage of this type of analysis, which – by including participant and item as random effects – allows to evaluate the effect of the main variables of interest (including age) by partialling out variability related to single items or single participants.
Concerning the inclusion and exclusion criteria, the reviewer is correct. This study had strict criteria to have comparable groups (except for the presence of aphasia). Our study aimed to investigate whether the adaptation of NAVS/NAT to the German language could identify grammatical deficits in patients with stroke-induced aphasia. Thus, we should limit the explanation of the variance on tests’ performance using a control group with comparable demographic, cognitive, and clinical characteristics. While some of the reported criteria were typically reported in other studies (such as sufficient cognitive abilities or right-handedness), or are intuitively essential (such as the limitation to native German speakers for adapting English tests into the German language), other factors, such as early bilingualism or music education at the professional level, for example, were selected with the intention of being methodologically strict and rigorous. Early bilinguals were excluded because bilingualism is considered a complicating factor for syntax processing, cognition, and the brain (Kroll et al. 2015). Music education and experience are known to induce brain-plasticity (Gaser et al. 2003, Münte et al. 2002, Schlaug 2001) and benefit speech perception (Musso et al. 2019). More specifically, it has been found that language and music grammar largely overlap (Musso et al. 2017). Therefore, musicians’ benefits in musical syntactic abilities could, in turn, enhance language syntactic computation. However, the exclusion of professional musicians or early bilinguals with aphasia does not reduce the sample of patients with aphasia in a relevant way.
More relevant were the limitations due to clinical criteria, which are, however, strongly recommended (as demonstrated by the comments of reviewer 3). As this aspect is very important, we explained it in the introduction p. 6, line 275
“Notably, it has been shown that demographic factors such as age (Antonenko et al., 2013) and education (Dąbrowska, 2012), and clinical variables such as lesion location, size, etiology (hemorrhagic vs. ischemic), stroke severity, and the presence of a new stroke, are factors that can predict language ability within patients with aphasia (Hope, Seghier, Leff, & Price, 2013; Lipson et al., 2005; Thye & Mirman, 2018; Yourganov, Smith, Fridriksson, & Rorden, 2015). In light of this evidence, we decided to control for several of these factors, both with our study protocol and with our data analysis.
A second aim of the study was therefore to identify to what extent the NAVS/NAT-G performance might relate with demographics and clinical variables as a prerequisite for further, more extensive studies for test standardization and validation “.
We also completely rewrote the section “Study design” pages 7-9.
Page 7. Much of the MoCA is verbal (incl. the abstraction task). Why was it used?
We used the MoCA to control concomitant cognitive impairments (Nasreddine et al., 2005, 101-102).
As the reviewer stated, MoCA is a verbal task. However, it does not mean that MoCA is not suitable for patients with anomic deficits. Wood et al. (ref. 102) found that the subtests for orientation and executive function were not compromised in patients with anomic deficits, and based on these results, they can be used to identify patients with mild dementia.
In the manuscript, we specified this point on page 7, line 360:
“For some stroke patients who were part of the large-scale protocol study in Freiburg, it was possible to report the MoCA test scores (Nasreddine et al., 2005) in the follow-up phase. This test is very sensitive for testing general cognitive abilities, even in patients with anomic deficits (Wood et al., 2020).”
Please report all effect sizes. I would also appreciate power calculations for the most important effects and p-values close to the sig. threshold. In my view the latter are too readily dismissed given the small samples.
We are sorry we have not reported the effect sizes in the original manuscript for all analysis. The revised version of the manuscript contains several tables that help summarize the results; effect sizes (i.e., R2) and power calculations are now provided for every main effect and regression model.
Most results showed sufficient effect size and statistical power for discussion. In particular, the effect of canonicity in the aphasic group and the results of the comparison between this patient group and the control groups, give some indication that the NAVS-G and NAT-G may have the potential to investigate grammatical competence in (German) stroke patients with aphasia, in particular those with minimal language impairments.
Please consider that even results with small to moderate effects are still relevant to report and discuss for comparing with the original version results and supporting the adaptation to the German language.
Moreover, the use of mixed-effects regression is particularly suited – compared to ANOVAs – for small samples, given that main effects and interactions are evaluated by taking into account item-related and participant-related reliability. This, together with our selection of carefully matched groups of participants with and without aphasia, ensures that the results reported in the manuscript are meaningful indicators of participants’ language impairments.
I cannot evaluate the discussion without the requested additions to the data reporting. However, the discussion about education and grammatical knnowledge should include Street & Dabrowska's work on the topic.
We thank the reviewer for the input, we have now included Street & Dabrowska's work in the discussion of the effect of education on syntactic operations:
“Street & Dabrowska’s documented that education influences the ability to produce nouns with a correct inflection (Dąbrowska, 2001, 2008), as well as the ability to process passive (Dąbrowska & Street, 2006) and implausible sentences (Street & Dąbrowska, 2010), and attributed this effect to the asymmetries in distributing these syntactic constructions in spoken and written discourse. Their data provide evidence that difficulties in less educated speakers lead to less experience with specific linguistic constructions and, hence, lack their syntactic well-entrenched schemas. Performance in these speakers improved dramatically after additional experience with the relevant construction, showing that the initial differences in test scores are attributable to differences in familiarity with specific linguistic constructions (Dąbrowska, 2012).”
Reviewer 2 Report
Brief summary
This manuscript reports a validation study of two aphasia tests, i.e. NAVS and NAT, for the German language employing three groups of participants, i.e. people with aphasia, non-aphasic patients with right-hemisphere damage, and control participants. The two tests focus on verb and sentence processing. Results showed that the German versions are able to differentiate well between people with and without aphasia. The authors conclude that the tests form a valid clinical instrument to test grammatical skills in both comprehension and production.
Evaluation
The manuscript is mostly well written and carefully reports all the findings of the test results regarding the three groups of participants. The manuscript is a bit lengthy, though, and the choice of language (mainly lexicon) is sometimes a bit odd (see my comments below). Based on the reported results, it seems as if the validity of the tests has been proven. I am not an expert on sentence processing. However, I do have a number of small comments regarding the literature.
Minor comments
p. 2, lines 69-71: The authors discuss Musso et al.’s study here. They should as well refer to two production studies here, i.e. Koester et al. (2011, NeuroImage) demonstrating a functional role of LIFG (incl. Broca’s area) with respect to processing and combination of morphological units, specifically in compounding, and Kepinska et al. (2018, NeuroImage) showing the role of Broca’s area (BA 44/45) regarding functional connectivity in novel grammar learning.
p. 3, lines 119-121: Regarding the (structural) connectivity of Broca’s area with other brain areas (see also Kepinska et al., 2018) and its role for syntactic processing, the authors should also refer to Heim et al. (2009, HBM) demonstrating a role of Broca’s area during the processing of lexico-syntactic features and specifically the selection of determiners.
p. 25, line 1021: “However, the proper role of the right hemisphere in syntactic processing is not clear.” -> please see Kepinska et al. (2018, NeuroImage) for the contribution of right hemispheric areas to syntactic processing.
references:
Heim, S., Friederici, A. D., Schiller, N. O., Rüschemeyer, S.-A., & Amunts, K. (2009). The determiner congruency effect in language production investigated with functional MRI. Human Brain Mapping, 30, 928-940.
Kepinska, O., De Rover, M., Caspers, J., & Schiller, N. O. (2018). Connectivity of the hippocampus and Broca’s area during acquisition of a novel grammar. NeuroImage, 165, 1-10.
Koester, D. & Schiller, N. O. (2011). The functional neuroanatomy of morphology in language production. NeuroImage, 55, 732-741.
Typo’s, etc.
general observation: Tables and Figures referred to in the text should be capitalized – the authors vary between upper and lower case letters when referring to Tables and Figures.
line 26: remove “preliminary”
line 164: “linguist models” -> “linguistic models”
line 196: “morph syntax” -> “morpho-syntax”
line 276: “participants” -> “Participants”
line 279: change the dot following “males” into a comma
line 309: “Broca aphasia” -> “Broca’s aphasia”
line 321: “flections error” -> “flection errors” (probably: “inflection errors”)
line 358: “data bank” -> “data base”
line 368: “MRT” -> “MRI”
line 371: add a parenthesis following “information”
line 442/443: “correct inflected” -> “correctly inflected”
line 453: remove one of the two commas following “action”
line 515: “postdoc” -> “post-hoc”
line 541: “figure” -> which figure?
line 567: the overall accuracy is given followed by another percentage – it is unclear what this second percentage refers to (I guess the variance, but this should be made explicit); dots instead of commas should be used for the decimals
line 568: same issue as in the previous line (see also lines 606 and 607)
line 618: remove dot following “p”
line 661: add a full stop after “277”
lines 691-699: commas instead of dots are used in the stats reports; please correct
line 712: remove the parenthesis following “r= .954”
line 726: “block wise” -> “block-wise”
line 727: “slightly significant” -> remove “slightly” (there is no such thing as slightly significant; either an effect is significant or not)
line 730: “into” -> “to”
line 764: “Inline” -> better: “In line with this”
line 790: “figure 4” -> “Figure 4”
line 887: “plane” -> “plan”
line 897: “normal” -> “healthy”
line 899: “tacking” -> “taking”
line 903: “was” -> “is”
line 909: “existed already in” -> “exist already at”
line 917: “older” -> better: “elderly”
line 958: “grey” -> in US English “gray” (as in line 920); the authors should stick to one spelling; most of the manuscript is according to US spelling
line 961: “Inline” -> see above (line 764)
line 971: “Inline” -> see above (line 764)
line 976: “as far as known” -> “as far as we know”
line 984: “less” -> “reduced”
line 998: “Do to” -> “Due to”
line 1016: “table 2” -> “Table 2”
line 1059: “lower” -> “fewer”
line 1066: “these” -> “our”
line 1068: “argument” -> “arguments”
line 1068: remove “and”
line 1069: “verbs” -> “arguments”
line 1083: “temporoparietooccipital” -> better: “temporo-parieto-occipital”
line 1107: remove “the”
line 1116: “even their speech was a "word salad"” -> colloquial language; please re-phrase; e.g. “even when their speech was complete incomprehensible”
line 1119: “unreal” -> “artificial”
line 1120: “an universal” -> “a universal”
Author Response
Brief summary
This manuscript reports a validation study of two aphasia tests, i.e. NAVS and NAT, for the German language employing three groups of participants, i.e. people with aphasia, non-aphasic patients with right-hemisphere damage, and control participants. The two tests focus on verb and sentence processing. Results showed that the German versions are able to differentiate well between people with and without aphasia. The authors conclude that the tests form a valid clinical instrument to test grammatical skills in both comprehension and production.
Evaluation
The manuscript is mostly well written and carefully reports all the findings of the test results regarding the three groups of participants. The manuscript is a bit lengthy, though, and the choice of language (mainly lexicon) is sometimes a bit odd (see my comments below). Based on the reported results, it seems as if the validity of the tests has been proven. I am not an expert on sentence processing. However, I do have a number of small comments regarding the literature.
Minor comments
- 2, lines 69-71: The authors discuss Musso et al.’s study here. They should as well refer to two production studies here, i.e. Koester et al. (2011, NeuroImage) demonstrating a functional role of LIFG (incl. Broca’s area) with respect to processing and combination of morphological units, specifically in compounding, and Kepinska et al. (2018, NeuroImage) showing the role of Broca’s area (BA 44/45) regarding functional connectivity in novel grammar learning.
We thank the reviewer for these suggestions; we have now included them on page 3 line 96 (N. 50 and 51)
- 3, lines 119-121: Regarding the (structural) connectivity of Broca’s area with other brain areas (see also Kepinska et al., 2018) and its role for syntactic processing, the authors should also refer to Heim et al. (2009, HBM) demonstrating a role of Broca’s area during the processing of lexico-syntactic features and specifically the selection of determiners.
We thank the reviewer for these suggestions; we included them on page 3 line 99 (N.54, 51)
- 25, line 1021: “However, the proper role of the right hemisphere in syntactic processing is not clear.” -> please see Kepinska et al. (2018, NeuroImage) for the contribution of right hemispheric areas to syntactic processing.
We thank the reviewer for these suggestions; we included them on pages 3 line 255 and 33 line 1276
General observation: Tables and Figures referred to in the text should be capitalized – the authors vary between upper and lower case letters when referring to Tables and Figures.
line 1016: “table 2” -> “Table 2”
We thank the reviewer; Tables and Figures have been revised.
line 26: remove “preliminary”
We rewrote the sentence (line 30)
“These findings indicated that the NAVS-G and NAT-G has potential for testing grammatical competence in (German) stroke patients”.
line 164: “linguist models” -> “linguistic models”
line 196: “morph syntax” -> “morpho-syntax”
line 276: “participants” -> “Participants”
line 279: change the dot following “males” into a comma
line 309: “Broca aphasia” -> “Broca’s aphasia”
line 321: “flections error” -> “flection errors” (probably: “inflection errors”)
line 358: “data bank” -> “data base”
line 368: “MRT” -> “MRI”
line 371: add a parenthesis following “information”
line 442/443: “correct inflected” -> “correctly inflected”
line 453: remove one of the two commas following “action”
line 515: “postdoc” -> “post-hoc”
line 727: “slightly significant” -> remove “slightly” (there is no such thing as slightly significant; either an effect is significant or not)
line 764: “Inline” -> better: “In line with this”
line 961: “Inline” -> see above (line 764)
line 971: “Inline” -> see above (line 764)
line 730: “into” -> “to”
line 1066: “these” -> “our”
We thank the reviewer. We have made all corrections in the revised manuscript.
line 541: “figure” -> which figure?
line 790: “figure 4” -> “Figure 4
These figures were removed.
line 567: the overall accuracy is given followed by another percentage – it is unclear what this second percentage refers to (I guess the variance, but this should be made explicit); dots instead of commas should be used for the decimals
line 568: same issue as in the previous line (see also lines 606 and 607)
line 618: remove dot following “p”
line 661: add a full stop after “277”
lines 691-699: commas instead of dots are used in the stats reports; please correct
line 712: remove the parenthesis following “r= .954”
The results are now completely rewritten. We used dots instead of commas and displayed standard deviation with ±. We removed dot after p.
line 726: “block wise” -> “block-wise”
line 887: “plane” -> “plan”
line 897: “normal” -> “healthy”
line 899: “tacking” -> “taking”
line 903: “was” -> “is”
line 909: “existed already in” -> “exist already at”
line 976: “as far as known” -> “as far as we know”
line 984: “less” -> “reduced”
line 998: “Do to” -> “Due to”
line 1059: “lower” -> “fewer”
line 1068: “argument” -> “arguments”
line 1068: remove “and”
line 1069: “verbs” -> “arguments”
line 1083: “temporoparietooccipital” -> better: “temporo-parieto-occipital”
line 1107: remove “the”
line 1116: “even their speech was a "word salad"” -> colloquial language; please re-phrase; e.g. “even when their speech was complete incomprehensible”
line 1119: “unreal” -> “artificial”line
1120: “an universal” -> “a universal”
All changes were made in the revised manuscript.
line 917: “older” -> better: “elderly”
We substituted older with elderly
line 958: “grey” -> in US English “gray” (as in line 920); the authors should stick to one spelling; most of the manuscript is according to US spelling
The reviewer is right; as we used US English we used gray.
Reviewer 3 Report
This paper reports on the adaptation of two existing English language tests of syntactic language function in aphasia to German, and on data from administration of this newly adapted version to 14 participants with aphasia due to left-hemisphere stroke, 24 healthy individuals without brain damage, and 15 participants with right hemisphere stroke.
The paper's primary strengths include (1) sophisticated application of linguistic theory to the clinical assessment of verb and sentence processing in aphasia, and (2) an excellent foundation in the literature on aphasic language impairments with a focus on impairments of verb and sentence processing and their treatment. I also very much appreciate that the study sample of persons with aphasia was not restricted to those diagnosed with Broca's aphasia. However, these strengths are undercut by substantial weaknesses in other areas, most prominently with respect to the lack of guiding theoretical frameworks for test development and investigation of test validity. Another limitation of the paper is the extremely small sample size, which will limit application of many of the recommendations that would follow from selection of an appropriate framework for evaluating the reliability and validity of the score estimates provided by the new test. Given the small sample size and its relatively restricted range of aphasia severity, all conclusions that depend on the data reported here should be considered and described as preliminary and in need of replication in a larger sample.
While the manuscript is particularly strong in its reporting of validity evidence based on test content and, to lesser extent, relations to other variables and some aspects of internal structure (Algina & Penfield, 2009; Messick, 1995), it is notably lacking in basic information about the item-level internal structure of the tests. I'm referring here to correlational analyses of the relationships among the items both within and across the various tests and subtests. At a more advanced stage of this work when a substantially larger participant sample is available, factor analysis or other latent variable modelling approaches should be used to evaluate whether the empirical data support the structure suggested by the various divisions by task and item content and the associated test and subtest score estimates. Examination of corrected item-total correlations for the relevant test and subtest scores could be undertaken for the present sample of individuals with aphasia, though with a sample size of 14, these estimates will be very imprecise. Relatedly, computation and interpretation of a total NAVS/NAT score should be supported, minimally, by adequate corrected subtest-to-total score correlations.
Also notably lacking is any consideration of the reliability or precision of score estimates, which can be considered another kind of validity evidence. Cronbach's alpha could be plausibly used to estimate the internal consistency reliability of the homogenous subtests (e.g., the VNT) but given that its assumptions of unidimensionality and tau-equivalence are unlikely to hold for NAVS/NAT total score, it likely cannot be appropriately applied in that case, and the small sample size precludes any robust testing of those assumptions. If multiple administrations of any of the tests to the same individuals over time intervals long enough to mitigate practice effects but short enough to limit the risk of actual underlying change in syntactic ability, test-retest reliability could and should be reported. Related to the issue of reliability, throughout the paper, the authors claim that certain individual participants demonstrated impairment on a given subtest. How were these decisions operationalized? Typically such claims about individual score estimates should be supported by calculation of the standard error of measurement (which in the classical test theory framework requires a reliability coefficient), construction of a confidence interval, and comparison of that confidence interval to a cut score based on a normative sample. Alternative procedures have also been developed (e.g., Crawford & Garthwaite, 2002)
The approach taken to constructing a proposed shortened version of the test is problematic, resulting in inflated estimates of agreement between the short and full forms, and likely, a short form with sub-optimal psychometric properties. Taking the second issue first, there are more useful methods for selecting items for inclusion in a short form than random selection from among items of a given type. The particulars depend on the psychometric framework being used (classical test theory or item response theory), but in either case they involve generally involve selecting items that are the most discriminating, i.e., those that are most strongly correlated with the underlying trait represented by the total (subtest) score and thus most effectively separate individuals along the score continuum, with the goal of producing a short form with the highest possible reliability. In the classical test theory approach this is accomplished by inspection of corrected item-total correlations and in item response theory this is done with reference to item information functions that are derived from a latent variable measurement model. Either approach is typically applied with substantially larger samples sizes than reported here. It might be possible to make some preliminary suggestions about short forms based on item-total correlations, but confidence intervals about the correlations should be inspected as well, because they will likely be quite wide and preliminary item reductions may or may not cross-validate well in a larger independent sample. In general, I would recommend waiting until data from a larger and more representative sample is available before taking steps to shorten a test of this type.
On the issue of the reported correlations between the various block-based short forms and the full versions, unless I misunderstand the study design, these estimates are inflated by the fact that a substantial proportion of the actual data are shared between the two versions of the various subtests. A more valid way of investigating the agreement between short and long forms would be to obtain independent administrations of them on two separate occasions. Alternatively, if only a single administration of the full test is available, one could, e.g., correlate scores from blocks 1-3 with scores estimated from blocks 4-5, and use reliability estimates for the two parts of the test to dis-attenuate the correlation estimate for measurement error (Osborne, 2002).
The inclusion of MRI data is not well integrated into the paper, and beyond the RH-LH distinction is not especially relevant to the main goals of the paper. Also, the supplemental materials refer to diffusion-weighted imaging, which is not referred to in the main manuscript.
Confidence intervals should be reported for all correlation estimates.
The supplemental materials indicate that a substantial portion of the AAT data were missing. This should be reported in the main manuscript, along with a clear description of the reasons for the missingness. Were these data plausibly missing at random, and could the missingness be biasing the results? The supplemental material provides some explanation of missingness in the acute stage but does not mention why data are missing in the more relevant chronic phase.
Although the manuscript is for the most part reasonably well written, it would benefit substantially from thorough editing by a native English speaker familiar with the content. Some suggestions for revisions along these lines are provided below, but these suggestions are by no means exhaustive.
Additional line-by-line comments are provided below:
p. 1, first paragraph: I would suggest avoiding citing Hauser if at all possible, given his history of scientific misconduct.
pps 2-3, lines 99-101: The claim in this sentence is too broadly stated. As written, it seems to apply to all research on aphasia, but the citations suggest that the authors intend to refer only to research into the syntactic or sentence processing abilities of persons with aphasia.
p. lines 110-113: Awkward and ungrammatical English here, suggest the following revision: However, this area does not work alone – Broca’s area operates preferentially together with other left perisylvian regions with which it interacts along the dorsal and ventral pathways – nor does it operate exclusively in the language domain [61]
p.3 line 119: The phrase 'Modern tracking studies' is mildly ambiguous, suggest 'Modern fiber tracking studies'
p.5 lines 250-260: These sentences seem out of place here in the final paragraphs of the introduction and would seem to be more appropriatly placed below in section 2.3 that describes the adaptation of the NAVS and NAT to German.
p. 6 line 305: How was the diagnosis of aphasia (I'm referring to aphasia vs. absence of aphasia, not to aphasia subtype or severity) operationalized, given that several participants were classified by the AAT ALLOC procedure as "minimal/none"?
p.7 line 314, incorrect English, 'Our patients, which..." should be 'Our patients, who...'
p.7 Table 1A: why do the stroke severity metrics have three values separated by slashes?
p.10, line 391: The abbreviated names of the NAVS subtests (and all other abbreviatons as well) should be written out in full at their first mention.
p.12, iine 473: The abbreviation VBT is not defined. I suspect that this is a typo for 'VNT' based on the context, but I am not sure. Assuming this sentence does in fact refer to the VNT, the scoring rule in this sentence seems to apply to close synonums for the target word, in which case I don't think labelling them as semantic paraphasias is necessrily appropriate. How was the decision about whether the production had the 'same meaning' operationalized? Would it be possible to construct an exhaustive list of alternative correct responses for each item?
p.13 lines 501-517: I am very skeptical of the decision to treat the results for the healthy control and right hemisphere stroke samples as a single undifferentiated group. I strongly suspect that the lack of statistical differences in the overall NAVS/NAT score and the NAT score are due to low power; post-hoc power calculation for the former suggests power = 0.31. While participants with RH stroke would not be expected to demonstrate prominent syntactic deficits, they are fundamentally different from the healthy control participants a priori due to the brain injury and small differences in performance on at least some components of the tests would be expected.
p.13 line 548: Figure 2 is mistakenly referred to as Figure 1.
p. 13 lines 548-550, and p.23 903-906: To what extent does the observed negative correlation between age and accuracy on SPPT OR peformance depend on combining the healthy control and RH stroke samples? Is this correlation present in either of those subsamples indivdually? Given that the RH stroke sample appears to be substantially older on average than the healthy control sample, poorer performance among the RH stroke participants could be driving this correlation with the two groups combined.
p.14 line 566: The authors use of the verb "preserved" here in the Results section seems inappropriate because it represents a inferential conclusion about underlying function, rather than a direct reporting of observed results. Also, it's not actually clear what is meant by 'Verb comprehension by verb argument structure was preserved,' and in any case, no comparison with the control group is reported until further down on p.19. It seems that the result that the authors intend draw attention to here is the finding that among participants with aphasia, verb comprehension performance was not affected by argument structure.
p.14 lines 567-568: The percentages reported here seem to have commas instead of period to indicate the decimals.
p.20 line 740: Figure 3 does not seem to address "agreement concerning verbs and sentence-picture matching tasks," and it is not clear what is meant by 'agreement concerning verbs" in this context. Verb naming performance?
p.20 line 748: This sentence cites Figure 3 but seems to refer to Figure 4.
p. 21 lines 774-796: I although I agree with the major claims advanced in this paragraph, I think that the first sentence is quite awkwardly phrased and I take issue with the verb 'proved'. It is difficult to imagine that any single study with a sample size of 14 could categorically 'prove' anything substantive about aphasia. Rather I think these data are consistent with the conclusion that assessment of verb argument stucture processing abilities is important, because thematic role assignment is important for succesful sentence production.
p. 22 line 858-861: The first sentence here comes very close to making the claim that the NAVS-G/NAT-G total score can be used to diagnose or detect aphasia in individual cases (this is echoed in line 870). The finding of a statistically significant difference between the groups is relevant and perhaps necessary, but not sufficient for supporting this claim. Appropriate support would involve, .e.g., a reciever operating characteristic curve analysis, identification of a cut score, and reporting of sensitivity and specificity estimates, all of which should be based on larger samples than is reported here.
p.22-23 lines 874-876: This sentence was initially difficult to parse, but I believe that the authors are claiming that the correlation between the NAVS-G/NAT-G total score and the categorical ranking of aphasia severity by the AAT was stronger than the correlation of the AAT Global score with the AAT severity ranking. If this is correct, the claim should be supported by reporting the related results in the Results section. As it currently stands, the relevant paragraphas of the Results section report only the correlations between the NAVS-G/NAT-G total score on the one hand and the AAT severity ranking and global scores on the other. The correlation between the latter two variables is not reported, and no statistical comparison to support the claim made in lines 874-876 is reported.
p.23 lines 895-900: This paragraph is very difficult to understand. Also, it's not clear whether the authors are using the phrase "syntactic competence' as a general synonym for syntactic ability, or whether they intend the more specific interpretation in the sense of syntactic competence vs. syntactic performance, in light of the historical debate about the extent to which aphasic language impairments represent loss of linguistic competence vs. reduction in the ability to consistently perform the cognitive operations required for language use. The same quesition about competence vs. performance applies to p 5 line 242.
p.24 lines 959-960: I think the word 'load' is superfluous or misused here. I suspect 'ability', 'span' might be more appropriate choices, or the word could simply be dropped.
p.24 line 966: I am not well versed in the literature on normal syntactic processing and I may well be wrong, but I am skeptical that this is the first study to find a correlation between correct production of OR sentences and educational attainment.
p.25 line 1016: It's not clear what table is meant by "S-table 2". I thought perhaps it meant table 2 in the supplementary materials, but this table does not include stroke severity data for the RH participants, as suggested by the reference.
p. 26 line 1033: The authors refer here and elsewhere to "non-statistical" computations. An explanation of this term for readers (like me) who are unfamiliar with it would be helpful.
Table S3: The figure caption mentions 6 AAT scores describing different aspects of spontaneous speech, but the table appears to include only 5.
I recommend using person-first language throughout, e.g., 'persons with aphasia' or 'participants with aphasia', instead of 'aphasic patients', and 'participants' instead of 'patients'
Several references to figures throughout the text of the results and discussion seem to be in error.
References
Algina, J. & Penfield, R.D. (2009) Classical test theory. In R.E. Millsap & Maydeu-Olivares (Eds.), The SAGE handbook of quantitative methods in psychology (pps. 93-122). London: SAGE Publications.
Crawford, J. R., & Garthwaite, P. H. (2002). Investigation of the single case in neuropsychology: confidence limits on the abnormality of test scores and test score differences. Neuropsychologia, 40(8), 1196-1208.
Messick, S. (1995). Validity of psychological assessment: Validation of inferences from persons' responses and performances as scientific inquiry into score meaning. American psychologist, 50(9), 741-749.
Osborne, J. W. (2002). Effect sizes and the disattenuation of correlation and regression coefficients: lessons from educational psychology. Practical Assessment, Research, and Evaluation, 8(1), 11.
Author Response
Reviewer 3
This paper reports on the adaptation of two existing English language tests of syntactic language function in aphasia to German, and on data from administration of this newly adapted version to 14 participants with aphasia due to left-hemisphere stroke, 24 healthy individuals without brain damage, and 15 participants with right hemisphere stroke.
The paper's primary strengths include (1) sophisticated application of linguistic theory to the clinical assessment of verb and sentence processing in aphasia, and (2) an excellent foundation in the literature on aphasic language impairments with a focus on impairments of verb and sentence processing and their treatment. I also very much appreciate that the study sample of persons with aphasia was not restricted to those diagnosed with Broca's aphasia.
However, these strengths are undercut by substantial weaknesses in other areas, most prominently with respect to the lack of guiding theoretical frameworks for test development and investigation of test validity.
We thank this reviewer for his/her comments. The main aim of this work was to prove the feasibility (and not the validity) of the two existing tests for German patients with aphasia only. An assessment for testing grammar in the German language is urgently necessary and this study indicates that the German adaptation of the NAVS/NAT may have the potential to investigate grammatical competence in (German) stroke patients with aphasia, even those with minimal language impairments. We are aware that this study is a preliminary investigation, a first step of the long process for standardizing this test battery.
In the first draft of the manuscript, we carried away with enthusiasm and some sentences were completely inappropriate. We have reworked the whole manuscript keeping this point - test of feasibility (and not the validity) - always very evident, starting with the title. We proposed to change from
“NAVS- and NAT-German: Testing grammar for diagnosis of aphasia.”
to
“German Language Adaptation of the NAVS (NAVS-G) and of the NAT (NAT-G): Testing grammar in aphasia.”
We have rewritten sentences as
“The current study aims to provide a valid, sensitive, and specific instrument for test-ing grammar competence in the German language” (page 5 line 241)
“Our study protocol was dictated by the need to check the validity, sensitivity, and specificity of these two novel tests for German patients with aphasia “ (page 9 line 353)
“This study presents an adaptation of the NAVS and NAT into the German language. Preliminary results in two heterogeneous groups of participants with chronic stroke-induced aphasia and without aphasia indicated that the combination of NAVS-G and NAT-G turns out to be a valid instrument for not only for testing grammar but also for detecting even the slightest language disturbances in native German-speaking stroke patients” (Page 20, line 730)
In
“The current study aimed to test the "feasibility "of these two English tests for assessment of grammar deficits in German patients with language impairments …” (page 5 line 213).
More specifically:
“The first aim of this study was to replicate the results obtained with the English version regarding the effects of verb-argument structure complexity (i.e., better accuracy for one-argument than for two- and three-argument verbs) and canonicity (i.e., better accuracy for canonical than for non-canonical sentences), in German patients with aphasia …” (page 5 line 239)…
"A second aim of the study was therefore to identify NAVS-G and NAT-G performance might relate with demographics and clinical variables as a prerequisite for further, more extensive studies of test standardization and validation" (page 6 line 263).
“This study presents the adaptation of the NAVS and NAT to the German language, documenting its ability to detect grammar deficits in (German) stroke patients with chronic mild to minimal language impairments. Moreover, the present research demonstrates that nonlinguistic factors such as age, education and stroke-related factors (such as stroke severity and stroke localization) contribute to explain the variability in test performance in different ways, in participants with and without aphasia" (page 31 line 958).
We are not sure what the reviewer mean by: “lack of guiding theoretical frameworks for test development and investigation of test validity”.
As the reviewer correctly assessed in the previous comment, our study did not develop, rather adapted already existing and used tests - the NAVS/NAT (developed in English)- to the German language. We followed the same theoretical frameworks as suggested by Thompson et al. in the NAVS/NAT manual and manuscripts [1,2]. In the manual of the original versions are reported the standardization procedure and psychometric data for test validity and reliability in 26 healthy participants and 103 patients with aphasia. We add this notion in the revised manuscript on page 31, line 967:
“The validity of the NAVS was documented by standardization procedures and psychometric data for test validity and reliability in the respective manual“.
Concerning the guiding theoretical frameworks, probably, the reviewer refers to the validity of testing framework of the Standards for Educational and Psychological Testing. It describes five types of validity evidence that can be evaluated to justify test score interpretation and use: (1) test content, (2) response processes of respondents and users, (3) internal structure of the assessment test, (4) relations to other variables and (5) consequences of testing, as related to validity. We agree with the reviewer that especially new tests should follow this important guideline.
However, as Hawkins et al. (2019) in their “systematic descriptive literature review of health literacy assessments” note: “The health sector is lacking a theoretically-driven framework for the development, testing and use of health assessments.” The authors of three tests in the German language structurally examining syntactic abilities - the "Komplexe Sätze" [72], "Sätze verstehen" [73] and "Sprachsystematisches Aphasiescreening (SAPS)" [74] do not refer to all these five points to justify the tests. Rohde et. al (2018) published a systematic review of 161 language tests. The conclusion was “No speech pathology test was found which reported diagnostic data for identifying aphasia in stroke populations. A diagnostically validated post-stroke aphasia test is needed.” They included the AAT also in their revision, although this test is a standard language test for patients with aphasia in Germany. One of the problems is that the health sector uses its own guideline, the guideline that we explicitly followed. These guidelines take in foreground other aspects (such as the size and location of infarct lesions, clinical conditions, the interference of comorbidity). We would like to see more exchange between the two fields.
We aim to continue to work with the NAVS/NAT-G to have a validated test. The shortcomings of the present study will be the goals of the next studies. Nevertheless, the NAVS-G and NAT-G we hope the actual results may provide a foundation for the use of the NAVS/NAT-G in the assessment of language impairments in individuals with aphasia and constitute the first step toward a proper standardization of the tests in a larger, more diverse, group of participants.
Another limitation of the paper is the extremely small sample size, which will limit application of many of the recommendations that would follow from selection of an appropriate framework for evaluating the reliability and validity of the score estimates provided by the new test. Given the small sample size and its relatively restricted range of aphasia severity, all conclusions that depend on the data reported here should be considered and described as preliminary and in need of replication in a larger sample.
The reviewer is right: We tested only 15 patients with aphasia but less variability of language impairments. For the standardization, it is desirable for normative groups to include at least 100 individuals (Franzen, 2003). However, several authors strongly suggest before standardization to perform a precedent study to check basic aspects such as the feasibility of the test and the main covariates, that the standardization should be controlled. For such a study, as this study is, 15 patients should be sufficient. The manuscript described the results of the adaptation and not of the standardization, and a future study is required. We repeatedly note this in the revised manuscript:
Page 35, line 1192
“In this respect, the NAVS-G and NAT-G aphasia battery could be a useful tool for identifying syntactic deficits in this patient group. For verifying this, the replication of the present results in a larger sample of participants is necessary. At this stage of the study, it was not our intention to provide a validation and standardization of the NAVS-G and NAT-G. Nevertheless, we do not expect these tests to be specific for the stroke aphasic patient even in the future validation phase. Rather, we expect that (and will test for) the usefulness of these tests for detecting syntactic deficits in other clinical populations, such as individuals with Alzheimer's disease or children with language impairments [36].
Page 36, line 1248
“The finding that syntactic competencies could relate to aging needs to be replicated in a large sample (and best, evaluated together with functional imaging measures such as resting-state fMRI or DTI), but it is still an essential finding in this study. Only a few language tests (the Token Test (TT) for example) take “age” into account for the validation process, i.e., introducing correction scores for this confounding parameter. Normative scores will allow us to distinguish aging sentence error from aphasia symptoms. Due to the increasing aging of the German population, the implications of these findings are relevant.”
Page 36, line 1299
“Future studies will need to replicate these findings, perhaps using rs-fMRI and DTI to better differentiate the effect of education from the effect of syntactic complexity.”
Page 37 line 1320
“These finding are novel and limited to 15 patients. More data are necessary to replicate then and to better understanding the role of the right hemisphere in syntax processing, currently still unclear.
Page 37 line 1341
Although these findings need to be replicated in a larger sample of subjects, they provide evidence that selective deficits in grammar are affected by factors that lie outside of “language proper” [36].
For the adaptation, our study reported:
- a good definition of the target population and control populations well matched due to
- selective inclusion and exclusion criteria
- accurate lesion mapping of the patients’ stroke
- clinical scores
- the variable “arteria cerebri media stroke”
- the control of external variables: age and education (Lezak et al., 2004; Mitrushina et al., 2005)
- A close relationship between AAT scores - a test of known validity and the NAVS/NAT-G (see Mitrushina et al., 2005) for concurrent validity.
- Data analysis is now performed using mixed-effects logistic regression analyses, which allows evaluation of the effect of the main variables of interest, in conjunction and in interaction with other variables, and better assessment of factors that contribute to variability in task performance.
Moreover, please consider
- The power of most analysis is over 80%
- Several language tests are examined in a small sample of subjects: The Communicative Effectiveness Index (CETI), widely used in our clinic - was examined in 11 patients with acute aphasia and 11 with chronic aphasia. Functional Communication Profile (FCP) [64] on 16 aphasic patients; Communication Activities for Daily Living (CADL-2) on 10 aphasic patients, Functional Assessment of Communication Skills for Adults (ASHA FACS) on 15 post-stroke chronic aphasia [80] and 15 healthy older people.
- Even results with small to moderate effects are still relevant to report and discuss for comparing with the original version results and supporting the adaptation to the German language.
While the manuscript is particularly strong in its reporting of validity evidence based on test content and, to lesser extent, relations to other variables and some aspects of internal structure (Algina & Penfield, 2009; Messick, 1995), it is notably lacking in basic information about the item-level internal structure of the tests. I'm referring here to correlational analyses of the relationships among the items both within and across the various tests and subtests. This work when a substantially larger participant sample is available, factor analysis or other latent variable modelling approaches should be used to evaluate whether the empirical data support the structure suggested by the various divisions by task and item content and the associated test and subtest score estimates. Examination of corrected item-total correlations for the relevant test and subtest scores could be undertaken for the present sample of individuals with aphasia, though with a sample size of 14, these estimates will be very imprecise. Relatedly, computation and interpretation of a total NAVS/NAT score should be supported, minimally, by adequate corrected subtest-to-total score correlations.
We thank the reviewer for this comment. Based on this and on other reviewers’ suggestions, we have completely re-analyzed all the data using logistic mixed-effects regression analyses, which allows a better evaluation of all the factors contributing to variability in task performance. We believe the results reported in the revised version of the manuscript are now even stronger than in the original version, and that they provide a better evaluation of the interaction of demographic and clinical variables with effects of verb argument structure and syntactic complexity.
In this revised version, we also provide correlations between performances on different subtests, which indicate that performance on most subtests correlate with overall accuracy on the NAVS-G and NAT-G.
As far as the item-level internal structure of the test, we agree with the reviewer that this is an important piece of information. Such analyses have been conducted for the original (English) version of the NAVS, and the results are reported in the test Manual (see below). Notably, for such analyses to be meaningful, we would need a larger and more diverse sample of participants. Therefore, we believe that such analyses should be undertaken for a future study of validity and standardization of the NAVS-G and NAT-G. However, we think such analyses are currently premature, as the goal of the present study was to provide a “proof of concept“ for the use of a German version of the NAVS-G and NAT-G as an assessment tool geared toward the quantification of grammatical impairments in individuals with aphasia.
Internal Reliability from NAVS (original)
To examine internal reliability, correlation analyses were conducted for all items across verb type within the Verb Comprehension Test (VCT; see Table 2)
For all items across sentence type within the Sentence Production Priming Test (SPPT; see Table 3) and the Sentence Comprehension Test (SCT; see Table 4).
Results of items analysis for SPPT-G
The results of corrected subtest-to-total score correlations were shown in the manuscript, Table 9. The analysis shows significant p-values (except VNT versus NAT), large Cohen's effect size - r - and acceptable statistical power at significance level (α) = .05
Also notably lacking is any consideration of the reliability or precision of score estimates, which can be considered another kind of validity evidence. Cronbach's alpha could be plausibly used to estimate the internal consistency reliability of the homogenous subtests (e.g., the VNT) but given that its assumptions of unidimensionality and tau-equivalence are unlikely to hold for NAVS/NAT total score, it likely cannot be appropriately applied in that case, and the small sample size precludes any robust testing of those assumptions. If multiple administrations of any of the tests to the same individuals over time intervals long enough to mitigate practice effects but short enough to limit the risk of actual underlying change in syntactic ability, test-retest reliability could and should be reported. Related to the issue of reliability, throughout the paper, the authors claim that certain individual participants demonstrated impairment on a given subtest. How were these decisions operationalized? Typically such claims about individual score estimates should be supported by calculation of the standard error of measurement (which in the classical test theory framework requires a reliability coefficient), construction of a confidence interval, and comparison of that confidence interval to a cut score based on a normative sample. Alternative procedures have also been developed (e.g., Crawford & Garthwaite, 2002)
As suggested by this reviewer, we performed the Cronbach’s Alpha statistic to estimate the reliability, or internal consistency. The Cronbach Alpha was .808
Item scale statistics |
||||
|
Scale mean if item deleted |
Scale variance if item deleted |
Corrected item total correlation |
Cronbach's Alpha if item deleted |
VBT |
424,9379% |
4681,938 |
,726 |
,782 |
VVT |
415,4682% |
5493,195 |
,479 |
,835 |
ASPT |
419,4949% |
4780,037 |
,572 |
,795 |
NAT |
438,7005% |
2302,684 |
,852 |
,733 |
SPT |
452,0339% |
2533,579 |
,870 |
,703 |
SVT |
424,8117% |
4384,443 |
,937 |
,753 |
We interpreted the results as promising internal consistency reliability. Again, we did not discuss this aspect in the manuscript because of the small sample of subjects, and because we did not intend to provide a standardization of the test.
The approach taken to constructing a proposed shortened version of the test is problematic, resulting in inflated estimates of agreement between the short and full forms, and likely, a short form with sub-optimal psychometric properties. Taking the second issue first, there are more useful methods for selecting items for inclusion in a short form than random selection from among items of a given type. The particulars depend on the psychometric framework being used (classical test theory or item response theory), but in either case they involve generally involve selecting items that are the most discriminating, i.e., those that are most strongly correlated with the underlying trait represented by the total (subtest) score and thus most effectively separate individuals along the score continuum, with the goal of producing a short form with the highest possible reliability. In the classical test theory approach this is accomplished by inspection of corrected item-total correlations and in item response theory this is done with reference to item information functions that are derived from a latent variable measurement model. Either approach is typically applied with substantially larger samples sizes than reported here. It might be possible to make some preliminary suggestions about short forms based on item-total correlations, but confidence intervals about the correlations should be inspected as well, because they will likely be quite wide and preliminary item reductions may or may not cross-validate well in a larger independent sample. In general, I would recommend waiting until data from a larger and more representative sample is available before taking steps to shorten a test of this type.
On the issue of the reported correlations between the various block-based short forms and the full versions, unless I misunderstand the study design, these estimates are inflated by the fact that a substantial proportion of the actual data are shared between the two versions of the various subtests. A more valid way of investigating the agreement between short and long forms would be to obtain independent administrations of them on two separate occasions. Alternatively, if only a single administration of the full test is available, one could, e.g., correlate scores from blocks 1-3 with scores estimated from blocks 4-5, and use reliability estimates for the two parts of the test to dis-attenuate the correlation estimate for measurement error (Osborne, 2002).
A detailed examination of syntax, which is the strength of these tests, should not be weakened by reducing the test material. Therefore, we removed this part of the results and discussion in the revised manuscript.
When we get the results of a larger sample of patients (with very severe aphasia and/or acute aphasia), we would like to verify to what extent a short version of NAVS-G and NAT-G may be used as a screening instrument.
The inclusion of MRI data is not well integrated into the paper, and beyond the RH-LH distinction is not especially relevant to the main goals of the paper. Also, the supplemental materials refer to diffusion-weighted imaging, which is not referred to in the main manuscript.
The inclusion of the MRI data is here limited to infarct mapping and the precise localization of the infarct. This information is very important to compare the two groups of patients: Infarct of the arteria cerebral media can cause different lesions. The figure showed that both groups had similar lesions, in terms of location and size.
Moreover, international clinical guidelines for stroke-induced rehabilitation recommended reporting all possible information of the patients for sharing data within different research groups (Rohde, Worrall, & Le Dorze, 2013; Y. Wang et al., 2019).
We have communicated that we performed DTI because it was part of our protocol. These data could help us to better interpret the data when we have a bigger sample of subjects.
Confidence intervals should be reported for all correlation estimates.
We thank the Reviewer for his/her comment. We have undertaken a new set of analyses using logistic mixed-effects regression analyses, and we have provided, whenever possible, beta estimates and standard error, as well as a measure of the goodness-of-fit of each model (i.e., the R2) and of statistical power for each predictor.
The supplemental materials indicate that a substantial portion of the AAT data were missing. This should be reported in the main manuscript, along with a clear description of the reasons for the missingness.
Were these data plausibly missing at random, and could the missingness be biasing the results? The supplemental material provides some explanation of missingness in the acute stage but does not mention why data are missing in the more relevant chronic phase.
There is a misunderstanding here. In the subsection “Study design”, we had explained that in clinical settings AAT is not automatically performed in a complete way for all patients with speech disorders. As indicated by the (German) manual, it should be carried out first the interview, then the Token Test and the writing subtests. If the patient shows a limited deficit in spontaneous speech and “normal” scores in these two subtests, it is not necessary to continue with the AAT assessment. The complete AAT is necessary only for patients with more important language impairments to define aphasia syndrome. Therefore, the data were not missing, but not carried out. In our study, we had to balance scientific goals with clinical constraints. The patients were examined during a visit in the stroke ambulance; they underwent many (neurological, neuropsychological, neuroradiological) tests. Moreover, patients with normal Token test and writing subtest, limited deficits in spontaneous speech do not profit by administration of a complete AAT, a clinical assessment that requires about one hour.
We have rewritten the caption in the Supplementary Table 3:
“The classification of aphasia severity is according to the AAT. In patients with minimal impairments in spontaneous speech and less than 4 errors in TT, the AAT was not carried out in all subsets. Patients N. 13 and 14 had a diagnosis of aphasia based on AAT spontaneous speech analysis only, because AAT's testing was not available. Patient N. 6 refused further testing. MoCA test could be performed only in patients of the university clinic Freiburg (patients displayed with * were external) and when the clinic profile was light. # Neither the AAT at the acute phase nor the MoCA in the chronic phase could be performed in these patients).
And paragraph (page 6) in hoping to be clearer.
“The presence of aphasia was first detected with the AAT compliant spontaneous speech analysis. It is, to some extent, the only method for longitudinal testing patients with severe aphasia in the acute phase [136, 137] or minimal deficits in the chronic phase [138, 139]. Nevertheless, reporting the individual longitudinal aphasia symptoms is an essential point for studying patients with residual aphasia [138, 139]. The AAT involves a structured interview covering four conversational topics (onset of illness, profession, family, and hobbies) of about 10-20 minutes. Spontaneous speech is evaluated on the basis of six levels of linguistic observation, with performance on each level being quantified on a 6-point scale (see Table 1). The scores’ assignment (0-5) on each level is defined by very specific and detailed criteria, so that an expert speech therapist assessed them. This neurolinguistic evaluation allows the degree of overall communicative impairment to be assessed, as well as the four standard clinical syndromes (for details see [67]). The AAT interview and its evaluation were carried out in both left- and right-hemispheric stroke patients for comparison.
Second, left-hemispheric stroke patients were tested using the remainder of the AAT in the acute phase (1-3 days after stroke), as this test is well standardized [140] and highly reliable [66]. According to the AAT manual, the binary diagnosis (aphasia yes/no) is determined by combining the evaluation of spontaneous speech with the Token Test (TT) subtest, which is also well standardized and highly sensitive for the diagnosis of aphasia [141, 142], and in few cases, the written language sub-test [140]. The assessment of aphasia severity is based on stanine norm data avail-able for each AAT subtest (see AAT manual). The complete AAT is performed for aphasia syndrome’s assignment, which is based on the AAT-ALLOC nonparamet-ric discriminant analysis [140]. During the acute phase six patients of this study could perform neither the AAT nor the TT because of the severity of aphasia (five patients) and one because of clinical instability. In the chronic phase, one patient refused further testing and for patients no. 13 and 14 diagnosis of aphasia was based on AAT spontaneous speech analysis only, because AAT's testing was not available. The combination of the AAT spontaneous speech, TT (and written lan-guage) subtests was sufficient for the classification to “no/residual aphasia“ cate-gory for four participants. This classification means that the patients’ clinical pic-ture did not lead to a specific syndrome, rather to broad language impairments al-ready evident in functional communication [143]. The complete AAT was administered to seven, more impaired patients, and the respective aphasia syndrome was reported (Table 1).”.
Although the manuscript is for the most part reasonably well written, it would benefit substantially from thorough editing by a native English speaker familiar with the content.
The revised version of the manuscript has been proofed by native speakers of English.
Some suggestions for revisions along these lines are provided below, but these suggestions are by no means exhaustive.
Additional line-by-line comments are provided below:
- 1, first paragraph: I would suggest avoiding citing Hauser if at all possible, given his history of scientific misconduct.
The cited manuscript - “The faculty of language: what is it, who has it, and how did it evolve? Science, 298(5598), 1569-1579. Hauser, M. D., Chomsky, N., & Fitch, W. T. (2002) - did not report the problematic comparative data; it is an update of generative grammar theory, which is particular interesting because Chomsky distances himself from the idea that grammar is a complete module human unique module.
However, we substituted the references in the submitted manuscript on pg 1:
“Notably, however, syntactic ability relates to organism-external factors including ecological, cultural and social environments, as well as broad sensory-motor operations (language perception and production) and conceptual-intentional computations such as (focused) attention and working memory in order to form thought and meaning (Chomsky 2005, Richards 2008)”.
pps 2-3, lines 99-101: The claim in this sentence is too broadly stated. As written, it seems to apply to all research on aphasia, but the citations suggest that the authors intend to refer only to research into the syntactic or sentence processing abilities of persons with aphasia.
We agree with the review and revised this sentence on pg. 1, line 82
Syntactic complexity affects verbs and sentences processing in stroke-induced aphasia - an acquired language disorder often resulting from lesions within the language network [36-39]. These data provide important evidence in favor of linguistic and psycholinguistic models for understanding both healthy and impaired sentence processing [9, 34, 36, 37, 40-47].
- lines 110-113: Awkward and ungrammatical English here, suggest the following revision: However, this area does not work alone – Broca’s area operates preferentially together with other left perisylvian regions with which it interacts along the dorsal and ventral pathways – nor does it operate exclusively in the language domain [61]
p.3 line 119: The phrase 'Modern tracking studies' is mildly ambiguous, suggest 'Modern fiber tracking studies'
We removed these sentences, as we shorten the introduction and now selectively focus on the importance for patients’ testing of valid assessments.
p.5 lines 250-260: These sentences seem out of place here in the final paragraphs of the introduction and would seem to be more appropriately placed below in section 2.3 that describes the adaptation of the NAVS and NAT to German.
We agree with the reviewer and moved, as suggested, the explanation of the differences between English and German language in the section 2.3
- 6 line 305: How was the diagnosis of aphasia (I'm referring to aphasia vs. absence of aphasia, not to aphasia subtype or severity) operationalized, given that several participants were classified by the AAT ALLOC procedure as "minimal/none"?
We apologize for the error; in the previously submitted manuscript at p. 7 line 310 we wrote: “In six patients in the chronic phase, AAT's constellation diagnosed "no aphasia/rest”, which was not correct and led to misunderstanding.
“Nine patients had a "residual aphasia" and presented language deficits in spontaneous speech [142], in the form of phonological or semantic paraphasias, syntactic or word-finding disorders (Table 1).
Please see also comment “The supplemental materials indicate that”
p.7 line 314, incorrect English, 'Our patients, which..." should be 'Our patients, who...'
This sentence was removed.
p.7 Table 1A: why do the stroke severity metrics have three values separated by slashes?
The three metrics were relative to NIHSS/mRS at admission, discharge, and at NAVS-G and NAT-G testing
We deleted the metrics for admission and discharge in the actual version.
p.10, line 391: The abbreviated names of the NAVS subtests (and all other abbreviatons as well) should be written out in full at their first mention.
The name of NAVS and its subsets were compared already on page 4, line 193
“In this light, a recently developed test, the Northwestern Assessment of Verbs and Sentences (NAVS) [2], is very promising in both design and results. The NAVS includes tests for (1) verb naming (i.e., the Verb Naming Test, VNT) ….”
p.12, iine 473: The abbreviation VBT is not defined. I suspect that this is a typo for 'VNT' based on the context, but I am not sure. Assuming this sentence does in fact refer to the VNT, the scoring rule in this sentence seems to apply to close synonums for the target word, in which case I don't think labelling them as semantic paraphasias is necessrily appropriate. How was the decision about whether the production had the 'same meaning' operationalized? Would it be possible to construct an exhaustive list of alternative correct responses for each item?
VBT is wrong, VNT is correct and we have corrected it respectively.
Healthy participants and right hemispheric stroke patients named all verbs correctly. Nevertheless, they used synonyms (now listed in supplementary materials “NAVS/NAT-G deficits):
- frieren statt zittern (freeze instead of shiver)
- ablegen/hinlegen/hinstellen instead of stellen (put down/put away instead of put
- parken instead of fahren (park instead of drive)
- herausziehen/herausholen instead of retten (pull out instead of rescue)
- schenken instead of geben (donate instead of give)
- kraulen instead of schwimmen (swim the crawl instead of swim)
- verfolgen, nachgehen (Dialect) instead of jagen (to pursue (dialect) instead of to hunt)
- kriechen instead of krabbeln (crawl instead of crawling)
- jaulen instead of heulen
- kehren oder schweifen (Dialekt) statt fegen (sweep or ramble (dialect) instead of sweep)
Patients did not use synonyms but paraphasia (schieben - to push - is not a synonym of ziehen - to pull).
p.13 lines 501-517: I am very skeptical of the decision to treat the results for the healthy control and right hemisphere stroke samples as a single undifferentiated group. I strongly suspect that the lack of statistical differences in the overall NAVS/NAT score and the NAT score are due to low power; post-hoc power calculation for the former suggests power = 0.31. While participants with RH stroke would not be expected to demonstrate prominent syntactic deficits, they are fundamentally different from the healthy control participants a priori due to the brain injury and small differences in performance on at least some components of the tests would be expected.
We agree with this comment, and we now report comparisons between the three groups of participants: healthy participants, individuals with aphasia and individuals without aphasia.
The difference between patients with aphasia and patients without aphasia were very clear at verb and sentence levels. More specifically, NAVS-G and NAT-G performance in patients with right hemispheric stroke (RHSP) did not statistically differ from HP, but did significantly differ from patients with aphasia, confirming the well accepted theory of left dominance for language (Vigneau, Mathieu, et al 2006, Hoffmann, et al. 202, Sauer et a. 2008, Glasser et al. 2998) and for grammar (Musso et al. 2003, Friederici et al. 2013).
However, RHSP showed some difficulty in objective relatives, subjective relative and objective questions. We discussed the role of the specific lesion localization on the right hemisphere for specific aspects of grammar processing. Thus, these findings are preliminary, but still important. Few language tests, such as the Token test (but not the well-standardized AAT) included the population of the right hemispheric stroke patients in their study-protocol. In line with functional imaging studies, our data suggest that it may also be important to have norm data for this population. It is also very interesting from a scientific point of view. The study takes place in Freiburg and we hope that, in the future, we can better define the difference between stroke patients with and without aphasia.
We have now added this aspect in the discussion, subsection 4.2.3 “The role of the right hemisphere on NAVS/NAT performance.”
p.13 line 548: Figure 2 is mistakenly referred to as Figure 1.
We apologize for this mistake.
- 13 lines 548-550, and p.23 903-906: To what extent does the observed negative correlation between age and accuracy on SPPT OR peformance depend on combining the healthy control and RH stroke samples? Is this correlation present in either of those subsamples indivdually? Given that the RH stroke sample appears to be substantially older on average than the healthy control sample, poorer performance among the RH stroke participants could be driving this correlation with the two groups combined.
We thank the Reviewer for this comment. We now report analyses for the two groups of individuals without aphasia separately, as well as between group analyses that evaluate the interaction between group and age in performance on the SPPT. Analyses showed that age affected sentence production on the SPPT for both groups of participants without aphasia, indicating better accuracy for younger (vs. older) participants in both groups. Notably, the interaction group*sentence type*age was significant; however planned comparisons within each participant group indicated no difference among sentence types in how age affected accuracy on the SPPT. Therefore, our discussion of the effects of age on sentence production has not focused exclusively on OR sentences.
p.14 line 566: The authors use of the verb "preserved" here in the Results section seems inappropriate because it represents a inferential conclusion about underlying function, rather than a direct reporting of observed results. Also, it's not actually clear what is meant by 'Verb comprehension by verb argument structure was preserved,' and in any case, no comparison with the control group is reported until further down on p.19. It seems that the result that the authors intend draw attention to here is the finding that among participants with aphasia, verb comprehension performance was not affected by argument structure.
We agree with the reviewer. We no longer not use the verb "preserve" (but “perform” or “carry out”) in the results session
.
p.14 lines 567-568: The percentages reported here seem to have commas instead of period to indicate the decimals.
We used dots instead of commas and we displayed the standard deviation with ±.
p.20 line 740: Figure 3 does not seem to address "agreement concerning verbs and sentence-picture matching tasks," and it is not clear what is meant by 'agreement concerning verbs" in this context. Verb naming performance?
p.20 line 748: This sentence cites Figure 3 but seems to refer to Figure 4.
We apologize for this error; we referred to the wrong Figure and Table.
- 21 lines 774-796: I although I agree with the major claims advanced in this paragraph, I think that the first sentence is quite awkwardly phrased and I take issue with the verb 'proved'. It is difficult to imagine that any single study with a sample size of 14 could categorically 'prove' anything substantive about aphasia. Rather I think these data are consistent with the conclusion that assessment of verb argument stucture processing abilities is important, because thematic role assignment is important for succesful sentence producti
We agree with the reviewer that these results are in no way definitive and should be replicated in a larger and more diverse sample of participants. However, we think this study provides the foundation for the use of the German NAVS/NAT in the assessment of grammatical production deficits in aphasia. Moreover, the study provides additional evidence that verb argument structure complexity and syntactic complexity affect grammatical production in aphasia, and extends findings derived from aphasic speakers of other languages to German and to individuals with minimal language disturbances.
Therefore, we agree with the reviewer and removed the term “proved” from the discussion of our data.
The sentence
“Our data proved that the identification of argument processing in this patient group is crucial as this disability, in turn, affects the production of active sentence”
was replaced with (page 33 line 1068)
“Our data supported the idea that testing verb argument processing in patients with aphasia is crucial as this impairment may also affect the production of active sentences (Thompson et al. 1997). .“
- 22 line 858-861: The first sentence here comes very close to making the claim that the NAVS-G/NAT-G total score can be used to diagnose or detect aphasia in individual cases (this is echoed in line 870). The finding of a statistically significant difference between the groups is relevant and perhaps necessary, but not sufficient for supporting this claim. Appropriate support would involve, .e.g., a reciever operating characteristic curve analysis, identification of a cut score, and reporting of sensitivity and specificity estimates, all of which should be based on larger samples than is reported here.
We are sorry, we had no intention of expressing that NAVS/VAT-G can diagnose aphasia. We indeed reported in the original manuscript: “The postulation that testing grammar may be sufficient for diagnosing aphasia needs to be proved with further studies and more participants, but, if confirmed, would have critical therapeutic repercussions. “
As suggested by the reviewer, we modified the sentences from:
“The comparison between the aphasic group and the control group demonstrated that the combination of the NAVS-/NAT-G is suited to identify syntactic deficits in a specific target population, one with aphasia. Patients with aphasia compared to the control group without a history performed significantly worse in all subtests except VCT-G and ASPT-G with monovalent verbs (Figure 4)”.
to (Page 35, line 1159)
“The comparison between the aphasic group and the control group indicates that the combination of the NAVS-G and NAT-G was able to identify language impairments in a target population, namely the stroke patients with aphasia “.
p.22-23 lines 874-876: This sentence was initially difficult to parse, but I believe that the authors are claiming that the correlation between the NAVS-G/NAT-G total score and the categorical ranking of aphasia severity by the AAT was stronger than the correlation of the AAT Global score with the AAT severity ranking. If this is correct, the claim should be supported by reporting the related results in the Results section. As it currently stands, the relevant paragraphas of the Results section report only the correlations between the NAVS-G/NAT-G total score on the one hand and the AAT severity ranking and global scores on the other. The correlation between the latter two variables is not reported, and no statistical comparison to support the claim made in lines 874-876 is reported.
This sentence led to a misunderstanding, as we did not claim that AAT severity correlated better than AAT-global scores with the performance in the NAVS/NAT-G. Therefore, we removed this sentence.
p.23 lines 895-900: This paragraph is very difficult to understand. Also, it's not clear whether the authors are using the phrase "syntactic competence' as a general synonym for syntactic ability, or whether they intend the more specific interpretation in the sense of syntactic competence vs. syntactic performance, in light of the historical debate about the extent to which aphasic language impairments represent loss of linguistic competence vs. reduction in the ability to consistently perform the cognitive operations required for language use. The same quesition about competence vs. performance applies to p 5 line 242.
We used "syntactic competence' as a general synonym for syntactic ability
As suggested we removed this paragraph.
p.24 lines 959-960: I think the word 'load' is superfluous or misused here. I suspect 'ability', 'span' might be more appropriate choices, or the word could simply be dropped.
We substituted ”-load” with “span”.
p.24 line 966: I am not well versed in the literature on normal syntactic processing and I may well be wrong, but I am skeptical that this is the first study to find a correlation between correct production of OR sentences and educational attainment.
There are only few studies investigating the relationship between “syntactic frequency” and education, but their topic was on subject relative, passive and nouns flection, but not on (OR) relative sentences. We added more references and have rewritten the subsection educational age.
p.25 line 1016: It's not clear what table is meant by "S-table 2". I thought perhaps it meant table 2 in the supplementary materials, but this table does not include stroke severity data for the RH participants, as suggested by the reference.
We removed this Table.
- 26 line 1033: The authors refer here and elsewhere to "non-statistical" computations. An explanation of this term for readers (like me) who are unfamiliar with it would be helpful.
“ Statistical" computations refers to the syntactic operations of “merge” and “move”, “ non-statistical” are the computations which according to generative grammar are “not proper grammatical”, as a part of the “broad language faculty in Heuser et al. (2002). However, we agree that the aspects of Universal grammar and “statistical” computations are not relevant for our manuscript.
Table S3: The figure caption mentions 6 AAT scores describing different aspects of spontaneous speech, but the table appears to include only 5.
The subtest spontaneous speech 4 was repeated twice
I recommend using person-first language throughout, e.g., 'persons with aphasia' or 'participants with aphasia', instead of 'aphasic patients', and 'participants' instead of ‚patients'
We thank the reviewer. We have substituted as suggested.
Several references to figures throughout the text of the results and discussion seem to be in error.
We thank the reviewer nd have accurately checked each reference to Figures and Tables.
References
Algina, J. & Penfield, R.D. (2009) Classical test theory. In R.E. Millsap & Maydeu-Olivares (Eds.), The SAGE handbook of quantitative methods in psychology (pps. 93-122). London: SAGE Publications.
Crawford, J. R., & Garthwaite, P. H. (2002). Investigation of the single case in neuropsychology: confidence limits on the abnormality of test scores and test score differences. Neuropsychologia, 40(8), 1196-1208.
Messick, S. (1995). Validity of psychological assessment: Validation of inferences from persons' responses and performances as scientific inquiry into score meaning. American psychologist, 50(9), 741-749.
Osborne, J. W. (2002). Effect sizes and the disattenuation of correlation and regression coefficients: lessons from educational psychology. Practical Assessment, Research, and Evaluation, 8(1), 11.

Round 2
Reviewer 3 Report
The authors' responsiveness to the issues raised in the first round of reviews is noted and appreciated. I find the authors' responses to almost all of the concerns I raised to be excellent or satisfactory, and in the few cases where disagreements remain, they are not germane to the revised manuscript. Overall I think the revised manuscript is much improved, and in particular I endorse the shift in analysis approach to focus on multilevel logistic regression models, as they provide a clearer basis for the inferences the authors wish to make. Also, I appreciate that the authors responded to my requests for item statistics including item-total correlations, and agree that, although they in general suggest that the items are appropriately discriminating, they do not belong in the revised manuscript.
I have essentially two remaining concerns:
- The description and reporting of the relatively large number of statistical models (the multilevel logistic regression models in particular) should be clearer. For each model, the fixed and random effects structure should be fully specified, and the results section or the supplementary materials should provide the complete table of parameter estimates for both the fixed and random effects. It should also be made clearer when results are coming from distinct models. For example, supplementary table 10 reports what appear to be post-hoc tests from multiple different overlapping models, but it's not clear. And Table 4 in the main manuscript mixes results from different models for both the VNT and ASPT and it's not clear which model the post-hoc results come from. Generally speaking, before reporting the results of post-hoc analyses, one should report the full results of the model those analyses are based on. Also, having a column labelled "chi-square (or z)" without making clear which values are which is not helpful to readers. Additionally, the methods section states that Wilcoxon tests were used in some cases because logistic regression models couldn't be fit due to low variability in the data, but it's not made clear in the results which cases these were.
- My second concern is that a substantial amount of awkward or unclear English remains in this revised version, as do many straightforward typos and other errors. I would suggest more careful and thorough proofreading and checking of the language by a native English speaker with content area knowledge. I have made detailed suggestions as comments in the attached pdf version of the revised manuscript and Word document version of the supplementary materials.

Author Response
Comments and Suggestions for Authors
The authors' responsiveness to the issues raised in the first round of reviews is noted and appreciated. I find the authors' responses to almost all of the concerns I raised to be excellent or satisfactory, and in the few cases where disagreements remain, they are not germane to the revised manuscript. Overall I think the revised manuscript is much improved, and in particular I endorse the shift in analysis approach to focus on multilevel logistic regression models, as they provide a clearer basis for the inferences the authors wish to make. Also, I appreciate that the authors responded to my requests for item statistics including item-total correlations, and agree that, although they in general suggest that the items are appropriately discriminating, they do not belong in the revised manuscript.
We thank the reviewer for all his/her suggestions.
I have essentially two remaining concerns:
The description and reporting of the relatively large number of statistical models (the multilevel logistic regression models in particular) should be clearer. For each model, the fixed and random effects structure should be fully specified. The results section or the supplementary materials should provide the complete table of parameter estimates for both the fixed and random effects. It should also be made clearer when results are coming from distinct models. For example, supplementary table 10 reports what appear to be post-hoc tests from multiple different overlapping models, but it's not clear. And Table 4 in the main manuscript mixes results from different models for both the VNT and ASPT and it's not clear which model the post-hoc results come from. Generally speaking, before reporting the results of post-hoc analyses, one should report the full results of the model those analyses are based on. Also, having a column labelled "chi-square (or z)" without making clear which values are which is not helpful to readers.
We agree with the Reviewer that we have reported the results of many models. The random and fixed effect structure was the same for all the subtests assessing verb processing (VNT and ASPT), with item and participant as random effects, and verb argument number (VAN), argument optionality (VAO) and verb type (VT) entered as fixed effects in separate analyses. For subtests of sentence processing (NAVS-G SPPT, NAVS-G SCT and NAT-G), the random and fixed effect structure was the same across subtests and included item and participant as random effects, and canonicity and sentence type entered as fixed effects in separate analyses. We have rewritten the paragraph on data analysis hoping that it is clearer than the previous version.
The greatest challenge was reporting the results, which should be precise, complete, and straightforward. The multi logistic regression uses different parameters for different effects. For example, for variables with three or more levels, Chi-squared statistics and p-values were derived from model comparisons (i.e., by comparing the model containing that variable to the model without that variable), z-values and p-values are provided for continuous variables or categorical variables with only two levels. Beta estimates and Standard errors are not provided for categorical variables with three or more levels because the predictor's overall significance was calculated by comparing models with and without it. Beta estimates and standard errors are provided for pairwise comparisons.
We modified all the tables as follows: For each subtest (with the exception of VCT, where accuracy was 100% across all groups and therefore no analyses were carried out) we reported a table summarizing the results of the models including a single predictor and the results of the adjusted models that yielded the best model fit, as well as the results of the between-groups analysis. To reduce the length of the Tables in the manuscript, we have reported results of post-hoc comparisons only when significant. The complete results are reported in the Supplementary materials. The same structure was used throughout the manuscript, to increase readability.
Additionally, the methods section states that Wilcoxon tests were used in some cases because logistic regression models couldn't be fit due to low variability in the data, but it's not made clear in the results which cases these were.
We thank the reviewer for the input, we have specified where we used Wilcoxon tests
My second concern is that a substantial amount of awkward or unclear English remains in this revised version, as do many straightforward typos and other errors. I would suggest more careful and thorough proofreading and checking of the language by a native English speaker with content area knowledge. I have made detailed suggestions as comments in the attached pdf version of the revised manuscript and Word document version of the supplementary materials.
We thank the reviewer for this comment. Although we were unable to see the Reviewer’s comments to the Supplementary materials (we have not received the word.document modified by reviewer), the current version of the manuscript has been proofed by several native speakers.
Minor points:
The table 1 column headed 'S6', which I assume is the one being referred to here, has 8 participants at '5', one participant at '1', and none at the minimum score of '0'. If this is not the column being referred to, it should be clarified so that readers can match the information provided in the text to the relevant table
We apologize for the error and thank the Reviewer for catching it. We amended as following: p. 8, line 366:
“Syntactic difficulties were, to some extent, detected by the AAT rating of syntactic structure in spontaneous speech based on the interview: eight patients had the maximal scores, while the others obtained lower scores due to production of short or very short sentences, and to omission/substitution of verb inflections.”
The end of this clause seems to contradict the beginning: "the full factorial model....the model was not full factorial". Is the issue here that number of arguments and optionality could not be fully crossed because optional 1-place verbs do not exist? If so, this should be stated more explicitly.
We thank the Reviewer for noticing what turned out to be a mistake. We have now clarified that verb argument numbers, verb optionality and verb type were entered as fixed effects in separate analyses.
Rev.3 wouldn't the more appropriate comparison here be obligatory 2- and 3-place verbs vs. optional 2- and 3-place verbs, excluding the 1-place verbs since they're all obligatory?
We agree with the Reviewer that ob1 verbs are all obligatory. For this reason, ob1 verbs were grouped with ob2 and ob3 (and compared to op2 and op3 together), consistently with the way Cho-Reyes & Thompson (2012) analyzed verb optionality in their manuscript. However, the Reviewer is correct, as comparing ob2/3 verbs versus op2/3 verbs allows to analyze the effect of optionality within verbs with the same number of arguments. This was achieved through the post-hoc analysis of verb type, which showed significantly better accuracy in op2/3 versus ob2/3.
„A second analysis was also carried out using verb type as a fixed effect with 5 levels (Ob 1-arg, Ob 2-arg, Ob 3-arg, Op 2-arg and Op 3-arg.)“: was this coded as a factor variable, and if so, what was the reference level?
We thank the Reviewer for raising this issue, which we have now clarified in the Data Analysis section. Categorical variables with 3 or more levels were all simple coded.
„Participant and Item were included as random effects in all analyses“. were these random effects included for only the intercept, or were random slopes also included, and if so, for which predictors?
Thank you for your comment. We have clarified that random intercepts for item and participant were introduced in all analyses.
The convention when plotting regression results is to put the dependent variable SPPT accuracy on the y-axis and the predictor variables on the x-axis.
We understand this may not be the Reviewer's preference, but we think the figure provides an accurate description of the effect.